# A geometric framework for momentum-based optimizers for low-rank training

**Steffen Schotthöfer**[*] **Timon Klein**[†] **and** **Jonas Kusch**[‡]

## Abstract

Low-rank pre-training and finetuning have recently emerged as promising techniques for reducing the computational and storage costs of large neural networks. Training low-rank parameterizations typically relies on conventional optimizers such as heavy ball momentum methods or Adam. In this work, we identify and analyze potential difficulties that these training methods encounter when used to train low-rank parameterizations of weights. In particular, we show that classical momentum methods can struggle to converge to a local optimum due to the geometry of the underlying optimization landscape. To address this, we introduce novel training strategies that combine dynamical low-rank approximation with momentum-based optimization, explicitly accounting for the intrinsic geometry of the parameter space. We validate our methods through numerical experiments, demonstrating stronger validation metrics at given parameter budgets.

## 1 Introduction

Deep learning models have achieved remarkable success across natural language processing and computer vision tasks, but their deployment remains computationally expensive due to the large number of trainable parameters. To address this, parameter-efficient strategies have been developed to reduce memory and compute requirements during training. Common approaches include sparsification [9, 22, 12], quantization [37, 6], and layer factorization. The latter has gained considerable attention for pre-training [36, 16, 28, 29, 41] and especially finetuning [15, 34, 42, 10, 43, 20, 27]. Layer factorization represents weights (or adapters) as low-rank matrices, allowing only the low-rank factors to be trained. This significantly reduces both memory usage and computational cost.

In the class of low-rank layer factorizations, one of the most popular methods is LoRA [15], which applied typical optimizers such as stochastic gradient descent (SGD) or Adaptive Moment Estimation (Adam) [17] directly to the low-rank factors. The combination of these optimizers with LoRA does not guarantee an optimal optimization trajectory [27, 42]. To overcome the former challenge, various improvements have been proposed: For example, LoRA+ [10] proposes the use of separate learning rates for different components of the low-rank decomposition, Dora [21] normalizes the factor matrices and introduces a magnitude parameter. Furthermore, to overcome the challenge of tuning the rank of the LoRA ansatz, AdaLoRA [42] adaptively allocating the parameter budget during training, by masking off rows and columns of the adapter matrices. Other low-rank methods focus entirely on the optimizer states: GaLore [43] projects full-rank weight gradients into a low-rank subspace to reduce the memory footprint of the optimizer state. Tensor-GaLore [7] generalizes this technique to high-order tensor-parameterized models, further improving efficiency for large-scale architectures. Adafactor [31] approximates the second-moment matrix using a low-rank decomposition of Adam with low-rank factors.

---

[*]Computer Science and Mathematics Division; Oak Ridge National Laboratory; Oak Ridge, TN 37831 USA; Mail correspondence: `schotthofers@ornl.gov`

[†]Department of Mathematics; Otto von Guericke University Magdeburg; 39106 Magdeburg; Germany

[‡]Scientific Computing; Norwegian University of Life Sciences; Drøbakveien 31, 1433 Ås; Norway

39th Conference on Neural Information Processing Systems (NeurIPS 2025).

A particularly promising class of layer factorization strategies is dynamical low-rank training (DLRT) which has been introduced in [28] and has since been used in various tasks [41, 29, 27, 5, 19]. DLRT projects the gradient flow dynamics onto the tangent space of the manifold of low-rank parameters, thereby achieving convergence guarantees to low-rank optimal weights [27]. However, these projections are inherently non-smooth, leading to ill-conditioned optimization landscapes and requiring smaller learning rates for stable training [28]. To ensure robust integration of such projections, DLRT constrains movement to flat subspaces within the low-rank manifold, enabling stable convergence to low-rank optima. The method is rank adaptive - using a basis augmentation and subsequent singular value-based truncation criterion to adapt the rank of the low-rank factorization. It further enables extensions to increase adversarial robustness of the compressed neural networks by enforcing orthonormality of the low-rank bases and projecting onto a well-conditioned manifold [26] or regularizing the condition number of the factor matrix [30].

Despite its efficiency, current DLRT approaches primarily rely on SGD, and it remains unclear how adaptive optimizers such as Adam [17] can be effectively applied. This is a significant limitation, as many state-of-the-art models depend on momentum-based optimizers like Adam and its variants for performance and stability. Therefore, the extension of DLRT to such optimization techniques is crucial.

To bridge this gap, we introduce a novel momentum-based optimization framework for low-rank pretraining and finetuning. The method integrates adaptive momentum techniques into the framework of dynamical low-rank training (DLRT), preserving the low-rank structure of model weights while enabling stable and efficient updates. This establishes a link between low-rank optimization and adaptive gradient methods, yielding both theoretical insights and practical improvements. Beyond the method derivation, we analyze why LoRA-style adapters—and low-rank parameterizations more broadly require momentum-based optimizers that are aware of the geometry of the low-rank parameter space, i.e., the underlying manifold structure. Naively applying standard optimizers such as heavy ball to low-rank parameterizations can produce updates that do not correspond to a gradient flow leading to a low-rank optimum. As a result, these methods may fail to converge to valid low-rank solutions. We show how DLRT can be adapted to approximate geometry-respecting gradient flows that consistently drive convergence toward low-rank optima. Together, these contributions lay a foundation for more robust and efficient training of large-scale models under low-rank constraints.

Compared to low-rank methods that act on the optimizer states only [43, 7, 31] we provide a holistic interpretation of a low-rank optimization algorithm that adaptively compresses the network weights, gradients and optimizer states simultaneously, achieving superior compression performance at high validation metrics. We remark that the method is directly extendable for tensor-valued neural networks using, e.g., low-rank Tucker factorization.

The paper is structured as follows: We first discuss limitations of naive momentum methods and show how to adapt the underlying gradient flow to achieve convergence to a low-rank optimum in Section 2. While the adapted gradient flow facilitates convergence, constructing robust numerical optimizers from it is challenging due to its inherent stiffness. We propose a low-rank heavy ball optimizer in Section 3 which integrates the adapted gradient flow robustly. In Section 4, we construct a fully low-rank Adam optimizer by leveraging insights gained in Section 3. In Section 5 we underline the efficiency of the proposed method through numerical experiments. In particular, we demonstrate fast convergence and superior validation accuracy at high compression levels for training from scratch, transfer learning, and low-rank finetuning of different neural network architectures and benchmarks.

## 2 Momentum-based low-rank training

We consider a low-rank neural network of the form

$$\mathcal{N}(x) = \sigma_L(U_L S_L V_L^\top z_{L-1}(x)) \,,$$

where $z_{L-1}(x)$ is defined recursively by

$$z_0(x) = x \in \mathbb{R}^{n_0} \,, \quad \text{and} \quad z_l(x) = \sigma_l(U_l S_l V_l^\top z_{l-1}(x)) \in \mathbb{R}^{n_l}, \quad \forall l = 1, \dots, L \,. \tag{1}$$

Here, the weight matrices are defined as $W_l := U_l S_l V_l^\top \in \mathbb{R}^{n_l \times n_{l-1}}$, where $U_l \in \mathbb{R}^{n_l \times r_l}$ and $V_l \in \mathbb{R}^{n_{l-1} \times r_l}$ are orthonormal low-rank factors, and $S_l \in \mathbb{R}^{r_l \times r_l}$ is the coefficient matrix. Thus, $W_l$ lies in the manifold of rank $r_l$ matrices which we denote by $\mathcal{M}_{r_l}$. Additionally, $\sigma_l$ represents

the activation function of layer $l$. For simplicity of notation, we do not consider biases, but a model with biases can always be expressed as Eq. (1) by folding biases into weights and creating an input dimension that is always one. Several methods have been proposed to train the low-rank weight matrices $W_l$ to minimize a given loss function $\mathcal{L}$. Among these, the simplest training rule is the steepest descent method, which, for a fixed[4] layer $l$ with weights $W = USV^\top \equiv W_l$ reads

$$U^{n+1} = U^n - \lambda \nabla_U \mathcal{L}^n, \quad S^{n+1} = S^n - \lambda \nabla_S \mathcal{L}^n, \quad V^{n+1} = V^n - \lambda \nabla_V \mathcal{L}^n, \qquad (2)$$

where $\lambda$ is the learning rate, the index $n$ denotes the training iteration, and we have used the shorthand notation $\mathcal{L}^n := \mathcal{L}(U^n S^n V^{n,\top})$. A more general framework that facilitates the numerical analysis and construction of novel numerical methods interprets the steepest descent method as an explicit Euler time discretization of the continuous gradient flow equations

$$\dot{U}(t) = -\nabla_U \mathcal{L}, \quad \dot{S}(t) = -\nabla_S \mathcal{L}, \quad \dot{V}(t) = -\nabla_V \mathcal{L},$$

where $\mathcal{L} := \mathcal{L}(U(t)S(t)V(t)^\top)$. Then, the steepest descent update equation for the individual factors can be obtained through a forward Euler discretization of the pseudo-time $t$. E.g., the update equation for $U$ can be retrieved through $\dot{U}(t_n) \approx \frac{1}{\lambda}(U^{n+1} - U^n)$ and $\mathcal{L} \approx \mathcal{L}^n$. While steepest descent methods offer a simple strategy to drive the parameters to a locally optimal point, one of the most widely adopted strategies for updating factorized parameters $W$, along with their factorized momentum terms $\mathcal{V} = U_\mathcal{V} S_\mathcal{V} V_\mathcal{V}^\top$, involves momentum-based optimization. For instance, in the case of the heavy ball method applied to all low-rank factors individually, e.g., in LoRA [15], the associated update equations take the form

$$U^{n+1} = U^n + \lambda U_\mathcal{V}^n, \quad U_\mathcal{V}^{n+1} = (1 - \lambda\gamma) U_\mathcal{V}^n - \lambda \nabla_U \mathcal{L}^n, \quad \nabla_U \mathcal{L}^n = (\nabla_W \mathcal{L}^n) V^n (S^n)^\top, \quad (3a)$$

$$V^{n+1} = V^n + \lambda V_\mathcal{V}^n, \quad V_\mathcal{V}^{n+1} = (1 - \lambda\gamma) V_\mathcal{V}^n - \lambda \nabla_V \mathcal{L}^n, \quad \nabla_V \mathcal{L}^n = (\nabla_W \mathcal{L}^n)^\top U^n S^n, \quad (3b)$$

$$S^{n+1} = S^n + \lambda S_\mathcal{V}^n, \quad S_\mathcal{V}^{n+1} = (1 - \lambda\gamma) S_\mathcal{V}^n - \lambda \nabla_S \mathcal{L}^n, \quad \nabla_S \mathcal{L}^n = (U^n)^\top (\nabla_W \mathcal{L}^n) V^n. \quad (3c)$$

The associated gradient flow equations are given by:

$$\dot{U} = U_\mathcal{V}, \quad \dot{U}_\mathcal{V} + \gamma U_\mathcal{V} + \nabla_U \mathcal{L} = 0, \quad \nabla_U \mathcal{L} = (\nabla_W \mathcal{L}) V S^\top, \qquad (4a)$$

$$\dot{V} = V_\mathcal{V}, \quad \dot{V}_\mathcal{V} + \gamma V_\mathcal{V} + \nabla_V \mathcal{L} = 0, \quad \nabla_V \mathcal{L} = (\nabla_W \mathcal{L})^\top U S, \qquad (4b)$$

$$\dot{S} = S_\mathcal{V}, \quad \dot{S}_\mathcal{V} + \gamma S_\mathcal{V} + \nabla_S \mathcal{L} = 0, \quad \nabla_S \mathcal{L} = U^\top \nabla_W \mathcal{L} V, \qquad (4c)$$

where $\gamma$ denotes the momentum decay parameter. While these equations are optimal when treating all low-rank factors in isolation, they do not account for the fact that factors change simultaneously. For example, Eq. (4a) is expected to drive the solution to an optimum, only if $S(t)$ and $V(t)$ remain constant in time. When accounting for the dynamics of all three low-rank factors, the resulting gradient flow for $W(t) = U(t)S(t)V(t)^\top$ takes the form

$$\dot{W} = U_\mathcal{V} S V^\top + U S_\mathcal{V} V^\top + U S V_\mathcal{V}^\top, \quad \text{and} \qquad (5a)$$

$$\begin{aligned} \dot{\mathcal{V}} + 3\gamma \mathcal{V} &= -\nabla_U \mathcal{L} S_\mathcal{V} V_\mathcal{V}^\top - U_\mathcal{V} \nabla_S \mathcal{L} V_\mathcal{V}^\top - U_\mathcal{V} S_\mathcal{V} \nabla_V \mathcal{L}^\top \\ &= -(\nabla_W \mathcal{L}) V S^\top S_\mathcal{V} V_\mathcal{V}^\top - U_\mathcal{V} U^\top \nabla_W \mathcal{L} V V_\mathcal{V}^\top - U_\mathcal{V} S_\mathcal{V} S^\top U^\top \nabla_W \mathcal{L} \\ &=: -\widehat{P}(W, \mathcal{V}) \nabla_W \mathcal{L}. \end{aligned} \qquad (5b)$$

This can easily be shown with the product rule, e.g., $\dot{W} = \dot{U} S V^\top + U \dot{S} V^\top + U S \dot{V}^\top$ and plugging in time derivatives from Eq. (4). These evolution equations for $W$ and $\mathcal{V}$ are fundamentally different from the momentum-based gradient flow equations of the full-rank problem

$$\dot{W}_{\text{full}} = \mathcal{V}_{\text{full}}, \quad \dot{\mathcal{V}}_{\text{full}} + \gamma \mathcal{V}_{\text{full}} = -\nabla_W \mathcal{L}(W_{\text{full}}). \qquad (6)$$

Indeed, a proper formulation of Eq. (6) that drives the weights of a heavy ball method to a local optimum while preserving the low-rank representation of $W$, requires that

$$\dot{W} = P(W)\mathcal{V}, \quad \dot{\mathcal{V}} + \gamma \mathcal{V} = -P(W)\nabla_W \mathcal{L}, \qquad (7)$$

where for $W = USV^\top$, the projector onto the tangent space is given by $P(W)Z := UU^\top Z(I - VV^\top) + ZVV^\top$, see [18, Lemma 4.1] and Figure 1 for geometric interpretation. Note that in

---

[4]We restrict the discussion to a single layer without loss of generality, following the arguments of [27, Appendix I].

abuse of notation, we have recycled $W$ and $\mathcal{V}$ here to denote the weights and momentum terms following Eq. (7) instead of the naive Eq. (4). Then, the time evolution will drive $W$ into a low-rank steady state $(W^\star, \mathcal{V}^\star)$ such that $P(W^\star)\nabla_W\mathcal{L}(W^\star) = 0$, see Theorem 1. This steady state thus fulfills the optimality condition of a local optimum, see, e.g. [25, Theorem 3.4]. Such a condition is not ensured by the simultaneous descent equations of (4) since, in general, $\dot{W} \neq P(W)\mathcal{V}$ and $\widehat{P}(W,\mathcal{V})\nabla_W\mathcal{L} \neq P(W)\nabla_W\mathcal{L}$, see Theorem 4. Therefore, training low-rank factors with conventional momentum methods does not necessarily ensure convergence to a low-rank optimum.

We aim to derive a numerical method that is consistent with the optimal gradient-flow equations (7). A central limitation of Eq. (7) is that it does not preserve the low-rank structure of the momentum term $\mathcal{V}$, thus leading to prohibitive computational costs and memory requirements. Instead, we aim to derive a method that fulfills

$$\dot{\mathcal{V}} + \gamma\mathcal{V} = -P(\mathcal{V})\nabla_W\mathcal{L} \qquad (8)$$

which preserves the low-rank structure of the momentum term. Indeed, the factorized solution of Eq. (8) fulfills (see Theorem 2)

$$\dot{U}_\mathcal{V} = -(I - U_\mathcal{V}U_\mathcal{V}^\top)\nabla_W\mathcal{L}V_\mathcal{V}S_\mathcal{V}^{-1}, \qquad (9a)$$

$$\dot{V}_\mathcal{V} = -(I - V_\mathcal{V}V_\mathcal{V}^\top)\nabla_W\mathcal{L}^\top U_\mathcal{V}S_\mathcal{V}^{-\top}, \qquad (9b)$$

$$\dot{S}_\mathcal{V} = -\gamma S_\mathcal{V} - U_\mathcal{V}^\top\nabla_W\mathcal{L}V_\mathcal{V}, \qquad (9c)$$

with initial condition $U(0) = U_\mathcal{V}(0)$ and $V(0) = V_\mathcal{V}(0)$. While this formulation and in particular Eq. (7) provide a good basis for constructing numerical methods, it also introduces the inverse terms $S_\mathcal{V}^{-1}$ and $S_\mathcal{V}^{-\top}$ on the right-hand side, rendering the system highly stiff, especially when these matrices are ill-conditioned. This stiffness can be treated through robust time integrators [2, 3] developed in the field of dynamical low-rank approximation [18] which have also been used for stochastic-gradient descent methods in dynamical low-rank training [28, 41, 29, 27, 13, 30]. In the following section, we formulate an algorithm that provably approximates the gradient flow of Eq. (9) by following ideas of [2]. It turns out that this method is a consistent approximation of the optimal gradient flow of Eq. (7) under mild assumptions.

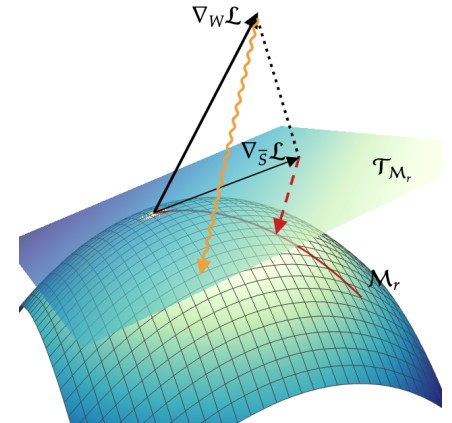

Figure 1: Geometric interpretation of Algorithm 1. We compute the parametrization of the tangent plane $\mathcal{T}_{\mathcal{M}_r}$. Then, we compute the projected gradient $\nabla_{\bar{S}}\mathcal{L}$ to construct the low-rank momentum update. The momentum optimizer is then applied to the low-rank weight coefficient $\widehat{S}$. Lastly, we retract the updated coefficients back onto the manifold $\mathcal{M}_r$. The interpretation of Algorithm 2 is analogous. LoRA-like methods do not employ orthogonal projections onto $\mathcal{T}_{\mathcal{M}_r}$, but instead map the full gradient $\nabla_W\mathcal{L}$ implicitly onto $\mathcal{M}_r$. The linear map (displayed as the wavy orange line) may map the gradient direction far away from the properly projected gradient flow, leading to suboptimal descent directions.

## 3 A low-rank heavy ball method

To approximate Eq. (7), we start with the first time step from $t_0 = 0$ to $t_1 = \lambda$ and note that by definition of the initial condition $U^0 := U(t_0) = U_\mathcal{V}(t_0)$ and $V^0 := V(t_0) = V_\mathcal{V}(t_0)$. Let us first construct an augmented basis to ensure that the range and co-range of $\mathcal{V}$ are fully spanned. To ensure robustness to small singular values, we introduce a change of variables and evolve the basis along $K_\mathcal{V}(t) = U_\mathcal{V}(t)S_\mathcal{V}(t)$ and $L_\mathcal{V}(t) = V_\mathcal{V}(t)S_\mathcal{V}(t)^\top$ for $t \in [t_0, t_1]$ while keeping $V^0 = V_\mathcal{V}(t_0)$ and $U^0 = U_\mathcal{V}(t_0)$ fixed, respectively [2]. Then, using the product rule and derivatives from Eq. (9), we get

$$\dot{K}_\mathcal{V}(t) = \dot{U}_\mathcal{V}(t)S_\mathcal{V}(t) + U_\mathcal{V}(t)\dot{S}_\mathcal{V}(t) \overset{(9)}{=} -\nabla_W\mathcal{L}(W(t))V^0, \qquad K_\mathcal{V}(t_0) = U^0S_v^0,$$

$$\dot{L}_\mathcal{V}(t) = \dot{V}_\mathcal{V}(t)S_\mathcal{V}(t)^\top + V_\mathcal{V}(t)\dot{S}_\mathcal{V}(t)^\top \overset{(9)}{=} -\nabla_W\mathcal{L}(W(t))^\top U^0, \quad L_\mathcal{V}(t_0) = V^0S_v^{0,\top}.$$

As no ill-conditioned $S_\mathcal{V}^{-1}$ terms affect the dynamics, one can use a forward Euler step to update $K_\mathcal{V}$ and $L_\mathcal{V}$ from $t_0$ to the next time step $t_1$. Thus, we get for $K_\mathcal{V}^n \approx K_\mathcal{V}(t_n)$, $L_\mathcal{V}^n \approx L_\mathcal{V}(t_n)$, and $W^n = W(t_n)$

$$K_\mathcal{V}^1 = K_\mathcal{V}^0 - \lambda\nabla_W\mathcal{L}(W^0)V^0, \qquad \text{with } K_\mathcal{V}^0 = U^0S_v^0,$$

$$L_\mathcal{V}^1 = L_\mathcal{V}^0 - \lambda\nabla_W\mathcal{L}(W^0)^\top U^0, \qquad \text{with } L_\mathcal{V}^0 = V^0S_v^{0,\top}.$$

**Algorithm 1:** Single iteration of the dynamical low-rank momentum method.
The functions `basis_augmentation`, and `truncation` are detailed in Algorithm 3 in the appendix.

---

**Input :** Initial orthonormal bases $U, V \in \mathbb{R}^{n \times r}$ and coefficients $S, S_{\mathcal{V}} \in \mathbb{R}^{r \times r}$;
$\tau$: singular value threshold for rank truncation;
$\lambda$: learning rate.

1  Evaluate $\mathcal{L}(USV^\top)$                                                   /* Forward evaluate */

2  $G_U \leftarrow \nabla_U \mathcal{L}(USV^\top); \; G_{\mathcal{V}} \leftarrow \nabla_V \mathcal{L}(USV^\top)$                    /* Backprop */

3  $\begin{cases} \widehat{U} \leftarrow \text{ basis\_augmentation}(U, G_U) \\ \widehat{V} \leftarrow \text{ basis\_augmentation}(V, G_{\mathcal{V}}) \end{cases}$                /* in parallel */

4  $\bar{S} \leftarrow \widehat{U}^\top USV^\top \widehat{V}; \; \bar{S}_{\mathcal{V}} \leftarrow \widehat{U}^\top US_{\mathcal{V}}V^\top \widehat{V}$

5  Evaluate $\mathcal{L}(\widehat{U}\bar{S}\widehat{V}^\top)$                                 /* Forward evaluate */

6  $G_S \leftarrow \nabla_{\bar{S}} \mathcal{L}(\widehat{U}\bar{S}\widehat{V}^\top)$                                  /* Backprop */

7  $\widehat{S}_{\mathcal{V}} \leftarrow (1-\gamma)\bar{S}_{\mathcal{V}} - \lambda G_S; \; \widehat{S} \leftarrow \bar{S} + \lambda \widehat{S}_{\mathcal{V}}$         /* coefficient update */

8  $U, S, V, S_{\mathcal{V}} \leftarrow \text{truncation}(\widehat{S}, \widehat{S}_{\mathcal{V}}, \widehat{U}, \widehat{V}; \tau)$

---

Denoting an orthonormalization algorithm like Gram-Schmidt as ortho, $K_{\mathcal{V}}^1$ and $L_{\mathcal{V}}^1$ are therefore spanned by

$$\widehat{U} = \text{ortho}(U^0, \nabla_W \mathcal{L}(W^0)V^0), \quad \widehat{V} = \text{ortho}(V^0, (\nabla_W \mathcal{L}(W^0))^\top U^0).$$

Following [40, Cor. 2.2], we note that $\nabla_U \mathcal{L} = \nabla_W \mathcal{L} VS^\top$ and $\nabla_V \mathcal{L} = (\nabla_W \mathcal{L})^\top US$, meaning that $\widehat{U}$ and $\widehat{V}$ can be rewritten as

$$\widehat{U} = \text{ortho}(U^0, \nabla_U \mathcal{L}(W^0)), \quad \widehat{V} = \text{ortho}(V^0, \nabla_V \mathcal{L}(W^0)).$$

We note here that this choice of the updated bases for $\mathcal{V}$ also approximates the updated parameters $W(t_1)$ of the equation $\dot{W} = \mathcal{V}$. With an implicit Euler time discretization, we have

$$W^1 = U^0 S^0 V^{0,\top} + \lambda \widehat{U} \widehat{S}_{\mathcal{V}}^1 \widehat{V}^\top = \widehat{U}(\widehat{U}^\top U^0 S^0 V^{0,\top} \widehat{V} + \lambda \widehat{S}_{\mathcal{V}}^1)\widehat{V}^\top =: \widehat{U}\widehat{S}^1 \widehat{V}^\top,$$

where $\widehat{S}^1 := \widehat{U}^\top U^0 S^0 V^{0,\top} \widehat{V} + \lambda \widehat{S}_{\mathcal{V}}^1$ and $\widehat{S}_{\mathcal{V}}^1$ is the time-updated coefficient matrix of $\mathcal{V}$ which we will derive in the following: Solving Eq. (9c) with fixed bases $\widehat{U}$ and $\widehat{V}$ yields

$$\dot{S}_{\mathcal{V}}(t) = -\gamma S_{\mathcal{V}}(t) - \widehat{U}^\top \nabla_W \mathcal{L}(W(t))\widehat{V} \qquad S_{\mathcal{V}}(t_0) = \widehat{U}^\top U^0 S_{\mathcal{V}}^0 V^{0,\top} \widehat{V}.$$

Here, we choose $S_{\mathcal{V}}(t_0)$ as the coefficient matrix $S_{\mathcal{V}}^0$ projected to the updated bases $\widehat{U}$ and $\widehat{V}$. This choice is crucial as it ensures the momentum term to be spanned with the updated basis. Using a forward Euler time discretization yields

$$\widehat{S}_{\mathcal{V}}^1 = (1-\gamma)\widehat{U}^\top U^0 S_{\mathcal{V}}^0 V^{0,\top} \widehat{V} - \lambda \widehat{U}^\top \nabla_W \mathcal{L}(U^0 S^0 V^{0,\top})\widehat{V}.$$

Let us note that with $\bar{S} := \widehat{U}^\top U^0 S^0 V^{0,\top} \widehat{V}$ we have

$$\widehat{U}^\top \nabla_W \mathcal{L}(U^0 S^0 V^{0,\top})\widehat{V} = \nabla_{\bar{S}} \mathcal{L}(\widehat{U}\bar{S}\widehat{V}^\top).$$

Thus, with $\bar{S}_{\mathcal{V}} := \widehat{U}^\top U^0 S_{\mathcal{V}}^0 V^{0,\top} \widehat{V}$, the final coefficient updates (including the update for $S$) are

$$\widehat{S}_{\mathcal{V}}^1 = (1-\gamma)\bar{S}_{\mathcal{V}} - \lambda \nabla_{\bar{S}} \mathcal{L}(\widehat{U}\bar{S}\widehat{V}^\top)$$
$$\widehat{S}^1 = \bar{S} + \lambda \widehat{S}_{\mathcal{V}}^1.$$

We note that the basis for $W$ and $\mathcal{V}$ remain identical after one time update, thus the above derivation holds for general time updates from $t_n$ to $t_{n+1}$. Since, by construction, the updated bases $\widehat{U}$ and $\widehat{V}$ have doubled in rank compared to $U^n$ and $V^n$, we perform a truncation step. The truncation can be performed back to the original rank $r$, or formulated with a relative truncation threshold for a given tolerance parameter $\vartheta = \tau \|\widehat{S}\|$ [2] or a rank budget [42], enabling a rank adaptive method.

**Algorithm 2:** Single iteration of the low-rank Adam method.
The functions `basis_augmentation` and `truncation` are detailed in 3 in the appendix.

---

**Input :** Initial orthonormal bases $U, V \in \mathbb{R}^{n \times r}$ and coefficients $S, S_\mathcal{V}, S_\mathcal{K} \in \mathbb{R}^{r \times r}$;
$\tau$: singular value threshold for rank truncation;
$\lambda$: learning rate;
$\beta_1, \beta_2$: Adam momentum parameters;
$\epsilon$: Small stability constant.

1   Evaluate $\mathcal{L}(USV^\top)$                                           `/* Forward evaluate */`

2   $G_U \leftarrow \nabla_U \mathcal{L}(USV^\top); \; G_V \leftarrow \nabla_V \mathcal{L}(USV^\top)$                       `/* Backprop */`

3   $\begin{cases} \widehat{U} \leftarrow \texttt{basis\_augmentation}(U, G_U) \\ \widehat{V} \leftarrow \texttt{basis\_augmentation}(V, G_V) \end{cases}$            `/* in parallel */`

4   $\bar{S} \leftarrow \widehat{U}^\top U S V^\top \widehat{V}, \; \bar{S}_\mathcal{V} \leftarrow \widehat{U}^\top U S_\mathcal{V} V^\top \widehat{V}, \; \bar{S}_\mathcal{K} \leftarrow \left( \widehat{U}^\top U \sqrt{S_\mathcal{K}} V^\top \widehat{V} \right)^2$

5   Evaluate $\mathcal{L}(\widehat{U} \bar{S} \widehat{V}^\top)$                                   `/* Forward evaluate */`

6   $G_S \leftarrow \nabla_{\bar{S}} \mathcal{L}(\widehat{U} \bar{S} \widehat{V}^\top)$                                `/* Backprop */`

7   $\widehat{S}_\mathcal{V} \leftarrow \beta_1 \bar{S}_\mathcal{V} + (1 - \beta_1) G_S$

8   $\widehat{S}_\mathcal{K} \leftarrow \beta_2 \bar{S}_\mathcal{K} + (1 - \beta_2)(G_S)^2$

    ▷ `Modifications for adaptive update`

9   $\check{S}_\mathcal{V} \leftarrow \frac{\widehat{S}_\mathcal{V}^n}{1 - \beta_1^n}, \; \check{S}_\mathcal{K} \leftarrow \frac{\widehat{S}_\mathcal{K}^n}{1 - \beta_2^n}$                           `/* Bias correction */`

10   $\widehat{S}^1 \leftarrow \bar{S} - \lambda \frac{\check{S}_\mathcal{V}}{\sqrt{\check{S}_\mathcal{K}} + \epsilon}$                        `/* Adaptive coefficient update */`

11   $U, S, V, S_\mathcal{V}, S_\mathcal{K} \leftarrow \texttt{truncation}(\widehat{S}, \widehat{S}_\mathcal{V}, \widehat{S}_\mathcal{K}, \widehat{U}, \widehat{V}; \tau)$

---

One step of the resulting method using a relative truncation threshold is summarized in Algorithm 1. A main distinction of this method from a naive application of a heavy ball method to DLRT [28] is that 1) our method uses the same bases for parameters and momentum terms, 2) our method does not use a momentum method to update the bases, but uses a classical basis augmentation instead, and 3) the method projects momentum terms onto the new bases after the basis update. These three choices ensure that the method approximates the optimal gradient flow of Eq. (7) independent of the condition number of $S^{-1}$ and $S_\mathcal{V}^{-1}$ when the truncation tolerance is sufficiently small, which we make rigorous in Theorem 3.

Theorem 3 shows that the low-rank momentum method produces solutions that are close to the solutions of full-rank (baseline) trained neural networks. Thus, we expect the validation accuracy of the trained low-rank networks to match the full-rank baseline. This is empirically confirmed in Table 1.

## 4   A low-rank Adam method

While heavy ball methods are the cornerstone of momentum-based optimization, they are commonly outperformed by momentum-based optimization methods that include stepsize control. Among these, perhaps the most popular method is Adaptive Moment Estimation (Adam), which was introduced in [17] and has significantly impacted the machine learning community; see Algorithm 4 for a reference formulation of Adam. While Adam exhibits superior performance, the non-linearities it introduces and a missing gradient flow formulation make a rigorous derivation of an extension to LoRA-type training difficult. In the following, we use the insights gained from the heavy ball method to construct a low-rank Adam optimizer. Adam's main distinction from heavy ball methods is the adaptive stepsize control, which is determined from the exponentially weighted moving average of the first and second moment of the gradient, denoted by $\mathcal{V}$ and $\mathcal{K}$.

A naive update of the exponentially weighted moving average of the first and second moment of the gradients with respect to the coefficients $S$, which we denote as $\mathcal{V}_S, \mathcal{K}_S \in \mathbb{R}^{2r \times 2r}$ would read as

$$\mathcal{V}_S^{n+1} = \beta_1 \mathcal{V}_S^n + (1 - \beta_1) \nabla_S \mathcal{L} \qquad \text{and} \qquad \mathcal{K}_S^{n+1} = \beta_2 \mathcal{K}_S^n + (1 - \beta_2)(\nabla_S \mathcal{L})^2 \,.$$

The bases $U, V$ of the weight factorization could be updated as in Algorithm 1. However, this naive approach does not account for the fact that $\mathcal{K}_S^n$ and $\mathcal{K}_S^{n+1}$, respectively $\mathcal{V}_S^n$ and $\mathcal{V}_S^{n+1}$ belong to different bases, which are updated between optimization steps by the augmentation-truncation mechanic.

Instead, we leverage the discussion in Section 3, and propose to project the previous weighted moving average to the updated bases. We note that through the construction of the low-rank bases, no step-size control is required to update $U$ and $V$ since the basis spans weights and moments at the old and current time steps and linear combinations in between. To facilitate the discussion, we denote the first low-rank moment as $S_{\mathcal{V}} \in \mathbb{R}^{r \times r}$, analogously to the momentum term in Algorithm 1 and note that the update of $S_{\mathcal{V}}$ can be performed using the strategy of Algorithm 1 with $\bar{S}_{\mathcal{V}} := \widehat{U}^\top U^0 S_{\mathcal{V}}^0 V^{0,\top} \widehat{V}$ and $\bar{S} := \widehat{U}^\top U^0 S^0 V^{0,\top} \widehat{V}$, i.e.,

$$\widehat{S}_{\mathcal{V}}^1 = \beta_1 \bar{S}_{\mathcal{V}} + (1 - \beta_1) \nabla_{\bar{S}} \mathcal{L}(\widehat{U} \bar{S} \widehat{V}^\top)$$

The dynamics of the second moment do not follow the gradient flow directly, but have non-linear dynamics depending on the square of the gradient. We remark that, the (full-rank) second moment $\hat{\mathcal{K}}$ is element-wise positive, which is important for taking the square root in the (full-rank) Adam update step $W^{n+1} = W^n - \lambda \frac{\hat{\mathcal{V}}}{\sqrt{\hat{\mathcal{K}} + \epsilon}}$. Denoting the second moment of the coefficient matrix by $S_{\mathcal{K}} \in \mathbb{R}^{r \times r}$, we propose a modified projection $\bar{S}_{\mathcal{K}} = \left( \widehat{U}^\top U^0 \sqrt{S_{\mathcal{K}}^0} V^{0,\top} \widehat{V} \right)^2$, which projects the element-wise square root of the second moment of the low-rank coefficient $S$ onto the new basis and subsequently squares the elements. This ensures element-wise positivity of the second low-rank moment. The augmented low-rank moment $\bar{S}_{\mathcal{K}}$ is subsequently updated with the standard Adam update scheme

$$\widehat{S}_{\mathcal{K}}^1 = \beta_2 \bar{S}_{\mathcal{K}} + (1 - \beta_2) \left( \nabla_{\bar{S}} \mathcal{L}(\widehat{U} \bar{S} \widehat{V}^\top) \right)^2 .$$

We apply the Adam bias correction at iteration $n$ to the low-rank moments, i.e.,

$$\check{S}_{\mathcal{V}} = \frac{\widehat{S}_{\mathcal{V}}^n}{1 - \beta_1^n}, \quad \text{and} \quad \check{S}_{\mathcal{K}} = \frac{\widehat{S}_{\mathcal{K}}^n}{1 - \beta_2^n} ,$$

and update steps for the coefficient $S$, i.e., $\widehat{S}^1 = \bar{S} - \lambda \frac{\check{S}_{\mathcal{V}}}{\sqrt{\check{S}_{\mathcal{K}} + \epsilon}}$.

The resulting method is summarized in Algorithm 2. We wish to remark that the extension to AdamW is straightforward: The regularization term is added to $\widehat{S}$ directly instead of being combined with the gradient for the moment updates.

**Computational and Memory Efficiency**

We briefly analyze the computational and memory efficiency of 1) the low-rank Heavy Ball method and 2) the low-rank Adam method. To update a matrix $W \in \mathbb{R}^{n \times n}$, the full-rank (baseline) Heavy Ball method requires $\mathcal{O}(3n^2)$ floats to store the weight $W$, its gradient $\nabla_W \mathcal{L}$, and the momentum $\mathcal{V}$. The computational cost is of the same order. In contrast, the low-rank Heavy Ball method (see Algorithm 1) requires $\mathcal{O}(2nr + r^2)$ floats to store the low-rank factorization $USV^\top$, and $\mathcal{O}(2nr + r^2)$ for the gradients $\nabla_U \mathcal{L}, \nabla_V \mathcal{L}, \nabla_S \mathcal{L}$. Since the bases $U$ and $V$ are shared between the weight and momentum, the momentum term requires only $\mathcal{O}(r^2)$ additional floats. The extra computational costs are $\mathcal{O}(nr^2)$ for orthonormalization during basis augmentation and $\mathcal{O}(r^3)$ for truncation, both negligible when $r \ll n$. Similarly, the full-rank Adam/AdamW method requires $\mathcal{O}(4n^2)$ floats to store $W, \nabla_W \mathcal{L}$, and the two momentum terms $\mathcal{V}, \mathcal{K}$, with comparable compute cost. The low-rank Adam method uses $\mathcal{O}(2nr + r^2)$ for $USV^\top$ and the same for the gradients. The two momentum terms add only $\mathcal{O}(2r^2)$ due to shared bases. The computational cost is computed analogously to that of the low-rank Heavy Ball method.

**Extension to Tensor-valued layers**

The proposed optimizer is directly extendable to tensor-valued neural network layers, e.g. convolutional layers, following the extension of the SGD-based DLRT method from matrices in [28] to tensors [41]. To that end, we remark that a, e.g. 2d, convolution can be formulated as an operation

on an order four tensor $W \in \mathbb{R}^{d_i \times d_o \times s_w \times s_h}$, where $d_i$ and $d_o$ are the channels of the input and output data, and $s_w, s_h$ is the width and height of the sliding window of the convolution acting on the spatial dimensions $s, h$ of the input image $x \in \mathbb{R}^{s \times h \times d_i}$. Consider a low-rank Tucker factorization of the weight, i.e. $W = C \times_{i=1}^{4} U_i$, with core tensor $C \in \mathbb{R}^{r_1 \times r_2 \times r_3 \times r_4}$ and $U_i \in \mathbb{R}^{n_i \times r_i}$ with $n_i \in \{d_i, d_o, s_w, s_h\}$. Using the extension of the DLRT method to Tucker tensors [40] and applying Algorithm 1, respectively Algorithm 2 to the tensor update yields the desired method.

# 5 Numerical Results

In the following, we showcase numerical experiments for low-rank transfer learning (Section 5.1), finetuning (Section 5.2), and pre-training tasks (Section 5.3). Details on training, data and additional experiments are listed in Section 10.

Our primary baseline is *full-rank training*, where all model parameters are updated without any structural constraints. We compare this against 1) the low-rank finetuning strategy introduced in Algorithm 2 2) naive application of DLRT without the projection of the momentum terms, and 3) simultaneous direct application of the

Table 1: UCM, Cifar10 and Cifar100 benchmark; Low-rank compressed transfer learning. Accuracy means and std. devs. of 10 stochastic trainings using AdamW. The LoRA ranks are set up to match the compression rate of the results of Algorithm 2. Algorithm 2 achieves higher accuracy at higher compression rates across all benchmarks, compared to DLRT[28] w/o momentum term projection and LoRA-based pretraining w/ momentum terms.

| | | UCM Data | | Cifar10 Data | | Cifar100 Data | |
|---|---|---|---|---|---|---|---|
| | | Acc [%] | c.r. [%] | Acc [%] | c.r. [%] | Acc [%] | c.r. [%] |
| VGG16 | Baseline | 94.40±0.72 | 0.0 | **89.82±0.45** | 0.0 | **65.21±0.37** | 0.0 |
| | Algorithm 2 | **94.61±0.35** | 95.84 | 89.49±0.58 | 95.30 | 64.58±0.46 | 95.54 |
| | DLRT w/o proj. | 89.32±0.93 | 93.56 | 85.01±0.28 | 94.84 | 60.48±0.27 | 98.71 |
| | LoRA pretrain | 90.64±2.27 | 93.57 | 88.43± 0.23 | 94.80 | 61.63±0.46 | 96.58 |
| VGG11 | Baseline | **94.23±0.71** | 0.0 | **88.34±0.49** | 0.0 | **63.13±0.41** | 0.0 |
| | Algorithm 2 | 93.70±0.71 | 94.89 | 88.13±0.56 | 95.13 | 60.84±0.40 | 95.08 |
| | DLRT w/o proj. | 88.23±0.90 | 90.35 | 81.98±0.25 | 97.08 | 61.59±0.25 | 95.99 |
| | LoRA pretrain | 90.14±2.56 | 94.72 | 86.63±0.29 | 94.57 | 59.54±0.40 | 94.78 |
| ViT-B.16 | Baseline | **96.72±0.36** | 0.0 | **95.42±0.35** | 0.0 | **90.34±0.44** | 0.0 |
| | Algorithm 2 | 96.38±0.60 | 86.7 | 95.39±0.41 | 83.42 | 88.48±0.53 | 75.38 |
| | DLRT w/o proj. | 78.94±0.50 | 84.91 | 91.95±0.50 | 84.95 | 75.09±0.53 | 75.83 |
| | LoRA pretrain | 86.54±2.91 | 84.94 | 94.10±0.56 | 80.78 | 76.76±0.53 | 74.86 |

Adam optimizer on the low-rank factors $U, S, V$, as typically done in LoRA [15], a low-rank adaptation technique originally designed for parameter-efficient finetuning of large transformers. For LoRA-based experiments, we calibrate the per-layer rank hyperparameters to match the overall *compression ratio* achieved by Algorithm 2, ensuring a fair comparison at fixed parameter budgets.

The compression ratio is defined as: c.r. $= \left(1 - \frac{\#\text{params low-rank model}}{\#\text{params baseline model}}\right) \times 100$.

## 5.1 Low-Rank Compressed Transfer Learning

**VGG16, VGG11, and ViT-B.16 on UCM/CIFAR-10/CIFAR-100** We evaluate performance across three network architectures, VGG16 and VGG11, and the ViT-B.16 Vision Transformer on UCM/CIFAR-10/CIFAR-100. The convolutional VGG11 and VGG16 networks are selected to validate the performance of Algorithm 2 on tensor-valued layers, here given by the convolutional layers. We use the low-rank Tucker tensor format to compress and train the convolutions; for details, we refer to [30, 40].

Table 2: DeBERTaV3-base finetuning on GLUE. We compare with full finetuning (Full FT), Houlsby adapter [14] (HAdapter), Pfeiffer adapter [23] (PAdapter), LoRA [15], AdaLoRA [42], GeoLoRA[27], DoRA [21], LoRA+[10], and Bitfit[39]. We report target metrics and computational performance (higher is better) for the median of 5 runs using different random seeds. Best results per dataset are shown in bold. Results for BitFit, HAdapter, and PAdapter were taken from [42]. "AdaLoRa matched" has the rank budget adapted to approximately match the final parameter count of Algorithm 2.

| Method (# Params) | SST-2 (Acc) | CoLA (Mcc) | QQP (F1) | QNLI (Acc) | RTE (Acc) | MRPC (Acc) | STS-B (Corr) | Mean |
|---|---|---|---|---|---|---|---|---|
| Full FT (184M) | 95.63 | 69.19 | 89.80 | 94.03 | 83.75 | 89.46 | **91.60** | 87.63 |
| BitFit (0.1M) | 94.84 | 66.96 | 84.95 | 92.24 | 78.70 | 87.75 | 91.35 | 85.25 |
| HAdapter (1.22M) | 95.53 | 68.64 | 89.27 | 94.11 | 84.48 | 89.95 | 91.48 | 87.63 |
| PAdapter (1.18M) | 95.61 | 68.77 | 89.40 | 94.29 | 85.20 | 89.46 | 91.54 | 87.75 |
| LoRA r=8 (1.33M) | 95.29 | 68.57 | 90.61 | 93.91 | 85.50 | 89.75 | 89.10 | 87.53 |
| LoRA+ r=8 (1.33M) | 95.37 | 69.22 | **90.82** | 93.96 | 85.50 | 89.55 | 88.07 | 87.49 |
| DoRA r=8 (1.33M) | 94.30 | 68.50 | 90.71 | **94.31** | 85.05 | 89.32 | 91.38 | 87.65 |
| AdaLoRA $r_f = 8$ (1.27M) | 95.64 | 68.76 | 90.65 | 94.11 | 86.00 | 89.44 | 91.41 | 88.00 |
| AdaLoRA, matched | 95.64 (1.27M) | 68.59 (1.07M) | 90.48 (0.72M) | 93.93 (0.72M) | 85.92 (1.16M) | 88.21 (0.74M) | 90.91(0.74M) | 87.66 (0.91M) |
| GeoLoRA | 95.98 (1.17M) | 69.03 (0.98M) | 90.53 (0.69M) | 94.23 (0.70M) | 85.93 (1.19M) | 90.10 (0.75M) | 91.58 (0.71M) | 88.19 (0.88M) |
| Algorithm 2 | **96.02** (1.11M) | **69.58** (1.01M) | 90.62 (0.76M) | 94.02 (0.70M) | **88.67** (1.19M) | **90.84** (0.76M) | 91.51(0.73M) | **88.75**(0.89M) |

We initialize all convolutional networks with PyTorch Imagenet1K weights and ViT-B.16 with Huggingface Imagenet21K weights. Stochasticity during training stems from randomized mini-batching. For each experiment, we report the mean performance across 10 independent training runs with different random seeds. We observe in Table 1 that Algorithm 2 matches the validation accuracy of the baseline network in most test-cases, and surpasses the baseline in e.g. UCM/VGG16 while achieving compression rates of up to $95\%$. We point out that a naive implementation of DLRT without the proposed adaptation of the optimizer states causes performance drops of 5 to 13%. LoRA also struggles to achieve high accuracy at the prescribed compression rates.

## 5.2 Low Rank Adaptation for Parameter Efficient Finetuning (PEFT)

**DeBERTaV3-base on GLUE**  We fine-tune the 183M parameter DeBERTaV3-base transformer model [11] on the GLUE benchmark suite [35]. The corresponding results are summarized in Table 2. Overall, Algorithm 2 consistently outperforms competing methods, especially other rank adaptive methods as GeoLoRA [27] and AdaLoRA [42], on most tasks, achieving stronger validation metrics. The required number of trainable parameters is substantially lower than the compared fixed-rank methods. The average score is higher than the reference methods, and the average parameter count for the finetuning tasks is lower than the next best method, which is GeoLoRA.

**Llama2 7b-chat-hf on BoolQ and PIQA**  We compare Algorithm 2 with LoRA on Llama-2-7b-chat-hf [33] across reasoning benchmarks, including BoolQ [4] and PIQA [1], as reported in Table 3. Inputs consist of either a passage–question pair or a standalone question with multiple-choice answers, and evaluation is based on answer accuracy. We also report wall-clock time on a single NVIDIA H100 GPU, showing negligible runtime overhead of Algorithm 2 over LoRA. Algorithm 2 outperforms LoRA configurations with matching initial rank and with rank chosen to approximately match the final parameter count.

Table 3: Llama2 7b-chat-hf [33] finetuning on reasoning datasets. We compare with LoRA [15] and report the best accuracy and the wall-time. The wall-time is reported for three epochs with batch size 12 and maximal sequence length of 640 tokens on a single NVIDIA H100.

| Method | BoolQ | | | PIQA | | |
|---|---|---|---|---|---|---|
| | c.r. [%] | Acc [%] | Wall-Time | c.r. [%] | Acc [%] | Wall-Time |
| Algorithm 2 | 99.73% | **84.09 %** | 186min | 99.75% | **76.77%** | 228min |
| LoRA (r=6) | 99.82% | 62.17 % | 173min | 99.82% | 52.18% | 225min |
| LoRA (r=10) | 99.70% | 62.17 % | 184min | 99.70% | 50.43% | 225min |

## 5.3 Low Rank Pretraining

**GPT2 on OpenWebText**  We pretrain Karpathy's reproduction[5] of the 124M-parameter GPT-2 model [24] from scratch on the OpenWebText dataset [8] using next-word prediction. As seen in Table 4 and Figure 4 , our method significantly outperforms LoRA pretraining (best validation loss 3.4642 vs. 4.8141), while incurring only a moderate increase relative to the full-rank baseline (3.4642 vs. 3.2313). Algorithm 2 achieves a compression rate of $60.61\%$, compared to $60.79\%$ for LoRA-Pretrain[6]. Thus, our approach enables substantial compression of GPT-2 (with the potential for reduced inference time), whereas LoRA yields a significant degradation in validation loss.

Table 4: Pretraining GPT-2 [24] reproduction (124M) from scratch on OpenWebText [8] for 15,000 iterations.

| Method | c.r. [%] | validation loss [%] |
|---|---|---|
| Baseline | 0 | 3.2313 |
| Algorithm 2 | 60.61% | **3.4642** |
| LoRA Pretrain | 60.79% | 7.0242 |

## 6  Conclusion

We introduced a principled and provably robust framework for momentum-based low-rank optimization that is both rank-adaptive and jointly compresses weights, gradients, and optimizer states. Our analysis reveals that the proposed method is resilient to the conditioning of the training problem, while maintaining fidelity to full-rank momentum trajectories. Through extensive experiments on pretraining, transfer learning, and finetuning, we demonstrate that our approach consistently achieves stronger generalization performance under tight parameter budgets, outperforming existing low-rank techniques. These results position our method as a strong foundation for efficient deep learning, offering a scalable and theoretically grounded alternative to full-rank training across diverse regimes. The accomplished faster convergence and higher compression of the optimizer states during training enable broader applications of machine learning on resource-constrained devices. Furthermore, the

---

[5]`https://github.com/karpathy/nanoGPT`

[6]We note that approximately 31% of the total parameters are in the encoding head of the model which is not compressed.

method is computationally efficient and scalable. These achievements also enhance computational and memory efficiency, positively impacting society.

## Funding Acknowledgements

The work of Steffen Schotthöfer is sponsored by the Applied Mathematics Progrm at the Office of Advanced Scientific Computing Research, U.S. Department of Energy, and performed at the Oak Ridge National Laboratory, which is managed by UT-Battelle, LLC under Contract No. DE-AC05-00OR22725 with the U.S. Department of Energy.

This manuscript has been authored by UT-Battelle, LLC under Contract No. DE-AC05-00OR22725 with the U.S. Department of Energy. The United States Government retains and the publisher, by accepting the article for publication, acknowledges that the United States Government retains a non-exclusive, paid-up, irrevocable, world-wide license to publish or reproduce the published form of this manuscript, or allow others to do so, for United States Government purposes. The Department of Energy will provide public access to these results of federally sponsored research in accordance with the DOE Public Access Plan(`http://energy.gov/downloads/doe-public-access-plan`).

This project has received funding from the European Regional Development Fund (grants timing-Matters and IntelAlgen) under the European Union's Horizon Europe Research and Innovation Program, from the German Research Foundation DFG within GRK 2297 'Mathematical Complexity Reduction', and from the German Federal Joint Committee (Grant 01VSF23017), which we gratefully acknowledge.

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

# 7  Notation

Table 5: Summary of notation used throughout the paper.

| Notation | Definition |
|---|---|
| ***Model & Training*** | |
| $W_l \in \mathbb{R}^{n_l \times n_{l-1}}$ | The weight matrix for layer $l$. |
| $\mathcal{L}$ | The loss function. |
| $\nabla_W \mathcal{L}$ | The gradient of the loss with respect to the full matrix $W$. |
| $\lambda$ | The learning rate of stochastic gradient descent. |
| ***Low-Rank Factorization*** | |
| $\mathcal{M}_r$ | The manifold of matrices of rank $r$. |
| $P(W)Z$ | Orthogonal projection of a matrix $Z$ onto the tangent space of $\mathcal{M}_r$ at $W$. |
| $W = USV^\top$ | Low-rank decomposition of $W$, where $U, V$ are orthonormal. |
| $U \in \mathbb{R}^{n_l \times r}, V \in \mathbb{R}^{n_{l-1} \times r}$ | Orthonormal basis matrices for the column and row spaces. |
| $S \in \mathbb{R}^{r \times r}$ | The core tensor or coefficient matrix. |
| $\nabla_U \mathcal{L}, \nabla_S \mathcal{L}, \nabla_V \mathcal{L}$ | Gradients with respect to the low-rank factors $U, S, V$. |
| ***Momentum Terms (Heavy Ball)*** | |
| $\mathcal{V}$ | The momentum term (a full-rank matrix). |
| $U_\mathcal{V} S_\mathcal{V} V_\mathcal{V}^\top$ | Low-rank decomposition of the momentum term. |
| $S_\mathcal{V} \in \mathbb{R}^{r \times r}$ | The coefficient matrix for the momentum term. |
| $\gamma$ | The momentum decay parameter. |
| ***Momentum Terms (Adam)*** | |
| $\mathcal{V}$ | The momentum term (a full-rank matrix). |
| $U_\mathcal{V} S_\mathcal{V} V_\mathcal{V}^\top$ | Low-rank decomposition of 1st moment term (moving average of gradients). |
| $S_\mathcal{V} \in \mathbb{R}^{r \times r}$ | Coefficient matrix for the 1st moment (moving average of gradients). |
| $\mathcal{K}$ | The momentum term (a full-rank matrix). |
| $U_\mathcal{K} S_\mathcal{K} V_\mathcal{K}^\top$ | Low-rank decomposition of the 2nd moment. |
| $S_\mathcal{K} \in \mathbb{R}^{r \times r}$ | Coefficient matrix for the 2nd moment (moving average of squared gradients). |
| $\beta_1, \beta_2$ | Exponential decay rates for the moment estimates. |
| ***Algorithm-Specific*** | |
| $\hat{U}, \hat{V}$ | Augmented bases after incorporating gradient information. |
| $\overline{S}$ | Coefficient matrix $S$ projected onto the augmented bases $\hat{U}, \hat{V}$. |
| $\overline{S}_\mathcal{V}, \overline{S}_\mathcal{K}$ | Momentum coefficients projected onto the new augmented bases. |
| $\hat{S}$ | The final updated coefficient matrix within a single optimization step. |
| $\tau, \vartheta$ | Relative and absolute truncation tolerance parameter for rank adaptation. |

# 8  Algorithms

We list the helper functions of Algorithm 1 and Algorithm 1 in Algorithm 3. The baseline (full-rank) Heavyball method is listed in algorithm 6 and the (full-rank) Adam method is listed in Algorithm 4. The naive application of the DLRT method [40] for Adam is listed in Algorithm 5. Note that the coefficient matrices of the momentum terms are not updated after augmentation and truncation. Analogously, the naive implementation of DLRT with momentum omits updating the momentum coefficient matrix. The implementation of LoRA with momentum or Adam simply applies Algorithm 4 to each of the LoRA factors without regard for the underlying manifold representation.

# 9  Numerical Analysis

**Theorem 1** (Convergence). *Let $W(t)$ be the solution of Eq. (7) and let $\mathcal{L}$ be bounded from below. Then, $W(t)$ converges to a $W^\star$ which fulfills the low-rank optimality condition*

$$P(W^\star)\nabla_W \mathcal{L}(W^\star) = 0 \, . \tag{10}$$

**Algorithm 3:** The functions `basis_augmentation`, and `truncation` of the used algorithms.

---

1 **def** `basis_augmentation`(*B: old basis*, $G_B$: *basis dynamics*)**:**

2     $\widehat{B} \leftarrow$ `ortho`($[G_B \mid B]$)        /\* orthonormalization, e.g. Gram-Schmidt \*/

3     return $\widehat{B}$

4 **def** `truncation`($\widehat{S}$: *augmented coefficient*, $\widehat{S}_{\mathcal{V}}$: *augmented momentum*, $\widehat{U}$: *augmented basis*, $\widehat{V}$: *augmented co-basis* )**:**

5     $P_{r_1}, \Sigma_{r_1}, Q_{r_1} \leftarrow$ truncated `svd`($\widehat{S}$) with threshold $\vartheta$ to new rank $r_1$

6     $U \leftarrow \widehat{U}P_{r_1}; V \leftarrow \widehat{V}Q_{r_1}$        /\* Basis update \*/

7     $S \leftarrow \Sigma_{r_1}; S_{\mathcal{V}} \leftarrow U^\top \widehat{U} \widehat{S}_{\mathcal{V}} \widehat{V}^\top V$        /\* Coefficient update \*/

8     return $U, S, V, S_{\mathcal{V}}$

9 **def** `truncation`($\widehat{S}$: *augmented coefficient*, $\widehat{S}_{\mathcal{V}}$: *augmented momentum*, $\widehat{S}_{\mathcal{K}}$: *augmented 2nd momentum*, $\widehat{U}$: *augmented basis*, $\widehat{V}$: *augmented co-basis* )**:**

10     $P_{r_1}, \Sigma_{r_1}, Q_{r_1} \leftarrow$ truncated `svd`($\widehat{S}$) with threshold $\vartheta$ to new rank $r_1$

11     $U \leftarrow \widehat{U}P_{r_1}; V \leftarrow \widehat{V}Q_{r_1}$        /\* Basis update \*/

12     $S \leftarrow \Sigma_{r_1}; S_{\mathcal{V}} \leftarrow U^\top \widehat{U} \widehat{S}_{\mathcal{V}} \widehat{V}^\top V; \widehat{S}_{\mathcal{K}} \leftarrow \left(U^\top \widehat{U} \sqrt{S_{\mathcal{K}}} \widehat{V}^\top V\right)^2$    /\* Coefficient update \*/

13     return $U, S, V, S_{\mathcal{V}}, S_{\mathcal{K}}$

14 **def** `truncation_naive`($\widehat{S}$: *augmented coefficient*, $\widehat{U}$: *augmented basis*, $\widehat{V}$: *augmented co-basis* )**:**

15     $P_{r_1}, \Sigma_{r_1}, Q_{r_1} \leftarrow$ truncated `svd`($\widehat{S}$) with threshold $\vartheta$ to new rank $r_1$

16     $U \leftarrow \widehat{U}P_{r_1}; V \leftarrow \widehat{V}Q_{r_1}$        /\* Basis update \*/

17     $S \leftarrow \Sigma_{r_1};$        /\* Coefficient update \*/

---

**Algorithm 4:** Single iteration of the (full-rank version of) Adam.

---

**Input :** $W \in \mathbb{R}^{n \times n}$: Weight matrix;
$\mathcal{V} \in \mathbb{R}^{n \times n}$: Initial 1st moment;
$\mathcal{K} \in \mathbb{R}^{n \times n}$: Initial 2nd moment;
$\lambda$: learning rate;
$\beta_1, \beta_2$: Adam momentum parameters;
$\epsilon > 0$: Small stability constant.

1 Evaluate $\mathcal{L}(W)$

2 $g \leftarrow \nabla_W \mathcal{L}(W)$        /\* Compute gradient \*/

3 $\mathcal{V} \leftarrow \beta_1 \mathcal{V} + (1 - \beta_1)g$        /\* 1st moment estimate \*/

4 $\mathcal{K} \leftarrow \beta_2 \mathcal{K} + (1 - \beta_2)g^2$        /\* 2nd moment estimate (element-wise square) \*/

5 $\hat{\mathcal{V}} \leftarrow \frac{\mathcal{V}}{1-\beta_1^t}, \quad \hat{\mathcal{K}} \leftarrow \frac{\mathcal{K}}{1-\beta_2^t}$        /\* Bias correction \*/

6 $W \leftarrow W - \lambda \frac{\hat{\mathcal{V}}}{\sqrt{\hat{\mathcal{K}}}+\epsilon}$        /\* Parameter update \*/

---

*Proof.* Let us define the energy as

$$E(t) := \mathcal{L}(W(t)) + \frac{1}{2}\|\mathcal{V}(t)\|^2 \,.$$

The time derivative is given by

$$\dot{E}(t) := \langle \nabla_W \mathcal{L}(W(t)), \dot{W}(t) \rangle + \langle \mathcal{V}(t), \dot{\mathcal{V}}(t) \rangle$$
$$= \langle \nabla_W \mathcal{L}(W(t)), P(W(t))\mathcal{V}(t) \rangle + \langle \mathcal{V}(t), -\gamma \mathcal{V}(t) - P(W(t))\nabla_W \mathcal{L}(W(t)) \rangle \,.$$

Since $P$ is self-adjoint this directly gives

$$\dot{E}(t) = \langle P(W(t))\nabla_W \mathcal{L}(W(t)), \mathcal{V}(t) \rangle + \langle \mathcal{V}(t), -\gamma \mathcal{V}(t) - P(W(t))\nabla_W \mathcal{L}(W(t)) \rangle$$
$$= -\gamma \|\mathcal{V}(t)\|^2 \,.$$

**Algorithm 5:** Single iteration of the naive low-rank Adam method.
The functions `basis_augmentation`, and `truncation_naive` are detailed in 3 in the appendix.

---

**Input:** Initial orthonormal bases $U, V \in \mathbb{R}^{n \times r}$ and coefficients $S, S_{\mathcal{V}}, S_{\mathcal{K}} \in \mathbb{R}^{r \times r}$;
$\tau$: singular value threshold for rank truncation;
$\lambda$: learning rate;
$\beta_1, \beta_2$: Adam momentum parameters;
$\epsilon$: Small stability constant.

**1** Evaluate $\mathcal{L}(USV^\top)$        /* Forward evaluate */

**2** $G_U \leftarrow \nabla_U \mathcal{L}(USV^\top)$; $G_V \leftarrow \nabla_V \mathcal{L}(USV^\top)$      /* Backprop */

**3** $\begin{cases} \widehat{U} \leftarrow \texttt{basis\_augmentation}(U, G_U) \\ \widehat{V} \leftarrow \texttt{basis\_augmentation}(V, G_V) \end{cases}$      /* in parallel */

**4** $\bar{S} \leftarrow \widehat{U}^\top U S V^\top \widehat{V}$

**5** Evaluate $\mathcal{L}(\widehat{U}\bar{S}\widehat{V}^\top)$        /* Forward evaluate */

**6** $G_S \leftarrow \nabla_{\bar{S}}\mathcal{L}(\widehat{U}\bar{S}\widehat{V}^\top)$        /* Backprop */

**7** $\widehat{S}_{\mathcal{V}} \leftarrow \beta_1 \bar{S}_{\mathcal{V}} + (1 - \beta_1)G_S$

**8** $\widehat{S}_{\mathcal{K}} \leftarrow \beta_2 \bar{S}_{\mathcal{K}} + (1 - \beta_2)(G_S)^2$
    ▷ Modifications for adaptive update

**9** $\check{S}_{\mathcal{V}} \leftarrow \frac{\widehat{S}_{\mathcal{V}}^n}{1 - \beta_1^n}$, $\check{S}_{\mathcal{K}} \leftarrow \frac{\widehat{S}_{\mathcal{K}}^n}{1 - \beta_2^n}$      /* Bias correction */

**10** $\widehat{S}^1 \leftarrow \bar{S} - \lambda \frac{\check{S}_{\mathcal{V}}}{\sqrt{\check{S}_{\mathcal{K}} + \epsilon}}$      /* Adaptive coefficient update */

**11** $U, S, V, S_{\mathcal{V}}, S_{\mathcal{K}} \leftarrow \texttt{truncation\_naive}(\widehat{S}, \widehat{U}, \widehat{V}; \tau)$

---

**Algorithm 6:** Single iteration of the (full-rank version of) the Heavy-Ball SGD method.

---

**Input:** Initial parameter vector $W \in \mathbb{R}^{n \times n}$;
$\mathcal{V}$: Initial velocity (momentum term);
Gradient $g = \nabla_W \mathcal{L}(W)$;
$\lambda$: learning rate;
$\gamma$: momentum coefficient.

**1** Evaluate $\mathcal{L}(W)$

**2** $g \leftarrow \nabla_W \mathcal{L}(W)$      /* Compute gradient */

**3** $\mathcal{V} \leftarrow (1 - \gamma)\mathcal{V} - \lambda g$      /* Update velocity */

**4** $W \leftarrow W + \lambda \mathcal{V}$      /* Parameter update */

---

Hence, if $\mathcal{L}$ is bounded from below, this means that $\lim_{t \to \infty} E(t) = E_\infty$ with $E_\infty$ finite and

$$E_\infty = E(0) - \gamma \int_0^\infty \|\mathcal{V}(t)\|^2 \, dt \,.$$

This implies that $\lim_{t \to \infty} \mathcal{V}(t) = 0$ and thus $\lim_{t \to \infty} \dot{W}(t) = \lim_{t \to \infty} P(W(t))\mathcal{V}(t) = 0$. Hence, since $\mathcal{V}(t), W(t)$ converge to a steady state and $\lim_{t \to \infty} \mathcal{V}(t) = 0$, the evolution equation for $\mathcal{V}$ gives $P(W(t))\nabla_W \mathcal{L}(W(t)) = 0$ as $t \to \infty$. $\qquad\qquad\qquad\qquad\qquad\qquad\square$

We can obtain a similar, but not equivalent, result when solving a low-rank gradient flow of the form (9) instead:

**Theorem 2** (Convergence of low-rank factors). *The low-rank gradient flow*

$$\dot{U}_{\mathcal{V}} = -(I - U_{\mathcal{V}}U_{\mathcal{V}}^\top)\nabla_W \mathcal{L}V_{\mathcal{V}}S_{\mathcal{V}}^{-1} \,, \tag{11a}$$

$$\dot{V}_{\mathcal{V}} = -(I - V_{\mathcal{V}}V_{\mathcal{V}}^\top)\nabla_W \mathcal{L}^\top U_{\mathcal{V}}S_{\mathcal{V}}^{-\top} \,, \tag{11b}$$

$$\dot{S}_{\mathcal{V}} = -\gamma S_{\mathcal{V}} - U_{\mathcal{V}}^\top \nabla_W \mathcal{L}V_{\mathcal{V}} \,, \tag{11c}$$

*fulfills*

$$\dot{\mathcal{V}} = -\gamma \mathcal{V} - P(\mathcal{V})\nabla_W \mathcal{L} \,.$$

*Proof.* By the product rule we have

$$\dot{\mathcal{V}} = \dot{U}_{\mathcal{V}} S_{\mathcal{V}} V_{\mathcal{V}}^\top + U_{\mathcal{V}} \dot{S}_{\mathcal{V}} V_{\mathcal{V}}^\top + U_{\mathcal{V}} S_{\mathcal{V}} \dot{V}_{\mathcal{V}}^\top$$
$$= -\gamma \mathcal{V} - (I - U_{\mathcal{V}} U_{\mathcal{V}}^\top) \nabla_W \mathcal{L} V_{\mathcal{V}} V_{\mathcal{V}}^\top - U_{\mathcal{V}} U_{\mathcal{V}}^\top \nabla_W \mathcal{L} V_{\mathcal{V}} V_{\mathcal{V}}^\top - U_{\mathcal{V}} U_{\mathcal{V}}^\top \nabla_W \mathcal{L}(I - V_{\mathcal{V}} V_{\mathcal{V}}^\top)$$
$$= -\gamma \mathcal{V} - P(\mathcal{V}) \nabla_W \mathcal{L}. \tag{12}$$

$\square$

**Theorem 3** (Error-bound). *For an integer $k$, let $t = k\lambda$. Let $W(t)$ be the solution of Eq. (7), and let $W_t^r$, $\mathcal{V}_t$ be the factorized low-rank solution after $k$ steps with Algorithm 1. Assume that for any $Z \in \mathcal{M}_r$ in a neighborhood of $W_t^r$, we have $\|(I - P(Z)) \nabla \mathcal{L}(Z)\| < \varepsilon$ and $\|\widehat{U}_t \widehat{U}_t^\top \mathcal{V}_t \widehat{V}_t \widehat{V}_t^\top - U_t U_t^\top \mathcal{V}_t V_t V_t^\top\| \le \widehat{\vartheta}$, where $\|\cdot\|$ denotes the Frobenius norm. Moreover, assume that the gradient is bounded and Lipschitz continuous. Then,*

$$\|W(t) - W_t^r\| \le c_1 \varepsilon + c_2 \lambda + c_3 \vartheta / \lambda + c_4 \widehat{\vartheta} / \lambda, \tag{13}$$

*where the constants $c_1$, $c_2$, $c_3$ are independent of singular values of $S^{-1}$ and $S_{\mathcal{V}}^{-1}$.*

*Proof.* We start by bounding the local error. That is, we assume that $W(t_0) = W_0^r$ and $\mathcal{V}(t_0) = \mathcal{V}_0^r$, where $\mathcal{V}_0^r$ is the momentum of the low-rank method. By definition of $\widehat{U}$ we have $(I - \widehat{U}\widehat{U}^\top) \mathcal{V}(t_0) = 0$ and thus

$$\|(I - \widehat{U}\widehat{U}^\top) \mathcal{V}(t)\| \le \int_{t_0}^t \|(I - \widehat{U}\widehat{U}^\top)(\gamma \mathcal{V}(s) + P(W(s)) \nabla_W \mathcal{L}(W(s)))\| \, ds.$$

Using the boundedness of normal components and a Taylor expansion around $t_0$ gives for $s \in [t_0, t_1]$

$$P(W(s)) \nabla_W \mathcal{L}(W(s))) = \nabla_W \mathcal{L}(W(s))) + O(\varepsilon) = \nabla_W \mathcal{L}(W(t_0)) + O(\lambda + \varepsilon)$$
$$= P(W(t_0)) \nabla_W \mathcal{L}(W(t_0)) + O(\lambda + \varepsilon). \tag{14}$$

Hence, with $\mathcal{V}(s) = \mathcal{V}(t_0) + O(\lambda + \varepsilon)$,

$$\|(I - \widehat{U}\widehat{U}^\top) \mathcal{V}(t)\| \le \lambda \|(I - \widehat{U}\widehat{U}^\top)(\gamma \mathcal{V}(t_0) + P(W(t_0)) \nabla_W \mathcal{L}(W(t_0)))\| + O(\lambda^2 + \lambda \varepsilon)$$
$$= \lambda \|(I - \widehat{U}\widehat{U}^\top) \nabla_W \mathcal{L}(W(t_0)) V_0 V_0^\top\| + O(\lambda^2 + \lambda \varepsilon).$$

By construction of $\widehat{U}$ we have $0 = (I - \widehat{U}\widehat{U}) \nabla_U \mathcal{L}(W(t_0)) = (I - \widehat{U}\widehat{U}) \nabla_W \mathcal{L}(W(t_0)) V_0$, hence

$$\|(I - \widehat{U}\widehat{U}^\top) \mathcal{V}(t)\| \le O(\lambda^2 + \lambda \varepsilon).$$

From this, we directly conclude

$$\|(I - \widehat{U}\widehat{U}^\top) W(t_1)\| = \|(I - \widehat{U}\widehat{U}^\top)(W(t_0) + \int_{t_0}^{t_1} \mathcal{V}(s) \, ds)\| = O(\lambda^3 + \lambda^2 \varepsilon).$$

An analogous derivation for the co-range gives

$$\|W(t_1) - \widehat{U}\widehat{U}^\top W(t_1) \widehat{V}\widehat{V}^\top\| \le \|(I - \widehat{U}\widehat{U}^\top) W(t_1)\| + \|W(t_1)(I - \widehat{V}\widehat{V}^\top)\|$$
$$= O(\lambda^3 + \lambda^2 \varepsilon).$$

Next, we need to bound

$$\|\widehat{U}\widehat{U}^\top W(t_1) \widehat{V}\widehat{V}^\top - \widehat{U}\widehat{S}^1 \widehat{V}^\top\| \le \|\widehat{U}^\top W(t_1) \widehat{V} - \widehat{S}^1\|. \tag{15}$$

We note that from Eq. (14) we have with $W_0 := W(t_0)$ and $\mathcal{V}_0 := \mathcal{V}(t_0)$

$$\widehat{U}^\top W(t_1) \widehat{V} = \widehat{U}^\top (W_0 + \lambda(1 - \gamma) \mathcal{V}_0 - \lambda^2 P(W_0) \nabla_W \mathcal{L}(W_0)) \widehat{V} + O(\lambda^2 + \lambda \varepsilon)$$
$$= \bar{S} - \lambda \gamma \bar{S}_{\mathcal{V}} - \lambda \widehat{U}^\top \nabla_W \mathcal{L}(W_0) \widehat{V} + O(\lambda^2 + \lambda \varepsilon),$$

where $\bar{S} = \widehat{U}^\top W_0 \widehat{V}$ and $\bar{S}_{\mathcal{V}} = \widehat{U}^\top \mathcal{V}_0 \widehat{V}$. By definition of the $S$-update of Algorithm 2 we have

$$\widehat{S}^1 = \bar{S} + \lambda(1 - \gamma) \bar{S}_{\mathcal{V}} - \lambda^2 \nabla_{\bar{S}} \mathcal{L}(\widehat{U} \bar{S} \widehat{V}^\top).$$

Thus, since $\nabla_{\bar{S}} \mathcal{L}(\widehat{U} \bar{S} \widehat{V}^\top) = U^\top \nabla_W \mathcal{L}(W_0) \widehat{V}$ we have $\|\widehat{U}^\top W(t_1) \widehat{V} - \widehat{S}^1\| = O(\lambda^2 + \lambda \varepsilon)$ and therefore the local error is bounded by

$$\|W(t_1) - W_1^r\| \le \|W(t_1) - \widehat{U}\widehat{U}^\top W(t_1) \widehat{V}\widehat{V}^\top\| + \|\widehat{U}\widehat{U}^\top W(t_1) \widehat{V}\widehat{V}^\top - \widehat{U}\widehat{S}^1 \widehat{V}^\top\|$$
$$= O(\lambda^2 + \lambda \varepsilon).$$

From the truncation tolerance $\vartheta$, the bound on the truncation of $\mathcal{V}$, and the stability of the exact flow, we can obtain the desired error bound for the global error using Lady Windermere's fan. $\square$

We remark, that we can always ensure that condition $\|\widehat{U}_t\widehat{U}_t^\top \mathcal{V}_t\widehat{V}_t\widehat{V}_t^\top - U_t U_t^\top \mathcal{V}_t V_t V_t^\top\| \leq \widehat{\vartheta}$, is fulfilled for a user determined $\widehat{\vartheta}$, e.g. $\widehat{\vartheta} = \vartheta$, by increasing the new rank $r_1$ in the truncation step of Algorithm 3 if necessary. However, since $\mathcal{V} \to 0$ when the method reaches a steady state, the effect of this error term is expected to be limited. We remark that the main motivation to present Theorem 3 is to rigorously demonstrate the robust treatment of stiff terms in the gradient flow (8). The main component in our construction of the algorithm, which removes these stiff terms in our error bound, is the construction of the augmented basis matrices $\widehat{U}$ and $\widehat{V}$.

**Theorem 4.** *The conventional low-rank gradient flow equations* (4) *can fail to converge to a point fulfilling* $P(W)\nabla_W\mathcal{L}(W) = 0$.

*Proof.* The potential lack of convergence can be proven with a counter-example. Consider a state where the momentum term is zero, i.e., $\mathcal{V} = 0$, by choosing the momentum factors as $U_\mathcal{V} = -U$, $S_\mathcal{V} = 0$, and $V_\mathcal{V} = V$. Now, consider any weight matrix $W$ that is not a low-rank optimum, meaning $P(W)\nabla_W\mathcal{L}(W) \neq 0$, but for which the naive update term in Eq. (5b) happens to be zero: $\hat{P}(W,\mathcal{V})\nabla_W\mathcal{L}(W) = 0$. In this scenario, the naive gradient flow equations would identify $(W, \mathcal{V})$ as a stationary point, since $\dot{W} = 0$ and $\dot{\mathcal{V}} = 0$. However, this point is not a valid low-rank optimum because the true projected gradient $P(W)\nabla_W\mathcal{L}(W)$ is non-zero. In contrast, $(W, \mathcal{V})$ is not a stationary point of (4) and the gradient flow (4) will provably drive the system to a state where $P(W)\nabla_W\mathcal{L}(W) = 0$. $\square$

We remark that a large class of matrices fulfills $P(W)\nabla_W\mathcal{L}(W) \neq 0$ and $\hat{P}(W,\mathcal{V})\nabla_W\mathcal{L}(W) = 0$. For instance, any $W = USV^\top$ where the gradient is non-zero in the range of $V$ but zero in the range of $U$ (i.e., $U^\top\nabla_W\mathcal{L}(W) = 0$ but $\nabla_W\mathcal{L}(W)V \neq 0$) would satisfy this. This can be easily verified: Since $\mathcal{V} = 0$, we have

$$\hat{P}(W,\mathcal{V})\nabla_W\mathcal{L}(W) = UU^\top\nabla_W\mathcal{L}(W)VV^\top = 0\,,$$

but

$$P(W)\nabla_W\mathcal{L}(W) = \nabla_W\mathcal{L}(W)VV^\top \neq 0\,.$$

## 10 Details to the numerical experiments of this work

It is important to note that the accuracy-vs-compression trade-off varies by application. While low-rank methods excel in finetuning and transfer learning tasks (sometimes even improving upon the baseline), pre-training a network from scratch on a complex dataset often involves balancing memory savings against a potential drop in accuracy.

### 10.1 ImageNet-1k, UCM and Cifar Benchmarks

#### 10.1.1 Network architecture and training details

In this paper, we use the pytorch implementation for neural network training. We take pretrained weights from the imagenet1k dataset as initialization, except for the long-term training study using ViT-small, which is randomly initialized. The data-loaded randomly samples a batch for each batch-update which is the only source of randomness in our training setup. Below is an overview of the used network architectures

- VGG16 is a deep convolutional neural network architecture that consists of 16 layers, including 13 convolutional layers and 3 fully connected layers.
- VGG11 is a convolutional neural network architecture similar to VGG16 but with fewer layers, consisting of 11 layers: 8 convolutional layers and 3 fully connected layers. It follows the same design principle as VGG16, using small 3×3 convolution filters and 2×2 max-pooling layers.
- ViT-B.16 is a Vision Transformer with $16 \times 16$ patch size, a deep learning architecture that leverages transformer models for image classification tasks.
- ViT-small is a compact vision transformer with patch size $8 \times 8$, and an embedding dimension of 512. The model comprises six attention layers, each equipped with two heads, followed by a ResNet block and a dropout layer.

Table 6: Training hyperparameters for the UCM, Cifar10, Cifar100 and ImageNet1k Benchmark. The first set hyperparameters apply to both DLRT and baseline training, and we train DLRT with the same hyperparameters as the full-rank baseline models. The second set of hyper-parameters is specific to DLRT. The DLRT hyperparameters are selected by an initial parameter sweep. We choose the DLRT truncation tolerance relative to the Frobenius norm of $\widehat{S}$, i.e. $\vartheta = \tau \|\widehat{S}\|_F$, as suggested in [28].

| Hyperparameter | VGG16 | VGG11 | ViT-B.16 | ViT-small | ViT-L.32 |
|---|---|---|---|---|---|
| Batch Size (UCM) | 16 | 16 | 16 | n/a | n.a. |
| Batch Size (Cifar10) | 128 | 128 | 128 | 256 | n.a. |
| Batch Size (Cifar100) | 128 | 128 | 128 | n.a | n.a. |
| Batch Size (ImageNet) | n.a | n.a | n.a | n.a | 256 |
| Learning Rate | 0.001 | 0.001 | 0.001 | 0.0001 | 0.001 |
| Number of Epochs (UCM, Cifar10) | 20 | 20 | 5 | 450 | n.a |
| Number of Epochs (Cifar100) | 30 | 30 | 20 | n.a | n.a |
| Number of Epochs (ImageNet1k) | n.a | n.a | n.a | n.a | 10 |
| L2 Regularization | 0 | 0 | 0.001 | 0.01 | 0.0001 |
| Optimizer | AdamW | AdamW | AdamW | Adam | AdamW |
| DLRT rel. truncation tolerance $\tau$ | 0.1 | 0.05 | 0.08 | 0.05 | 0.013 |
| Coefficient Steps $s_*$ | 10 | 10 | 10 | 75 | 75 |
| Initial Rank | 150 | 150 | 150 | 200 | 200 |
| Parameters | 138M | 132M | 86M | 50M | 304M |

- ViT-L.32 is a Vision Transformer with 32x32 patch size, a deep learning architecture that leverages transformer models for image classification tasks. We use the Imagenet21k weights from the huggingface endpoint google/vit-large-patch32-224-in21k as weight initialization.

The full training setup is described in Table 6. We train DLRT with the same hyperparameters as the full-rank baseline models. It is known [27] that DLRT methods are robust w.r.t. common hyperparameters as learning rate, and batch-size, and initial rank. The truncation tolerance $\tau$ is chosen per an initial parameter study. These values are are similar to default values reported in recent literature [29, 30, 32]. In general, there is a trade-off between target compression ratio and accuracy, as illustrated e.g. in [28] for matrix-valued and [32] for tensor-valued (CNN) layers.

### 10.1.2 UCM Data

The UC Merced (UCM) Land Use Dataset [38] is a standard benchmark in remote sensing and computer vision. It consists of 2,100 high-resolution aerial RGB images, each of size $256 \times 256$ pixels, organized into 21 land use classes with 100 images per class.

We normalize the training and validation data using channel-wise means $[0.485, 0.456, 0.406]$ and standard deviations $[0.229, 0.224, 0.225]$. Convolutional neural networks (CNNs) are applied directly to the original $256 \times 256$ image resolution. For the Vision Transformer (ViT), the input images are resized to $224 \times 224$ pixels within the data pipeline.

### 10.1.3 CIFAR-10 Data

The CIFAR-10 dataset comprises 60,000 RGB images of size $32 \times 32$ pixels, uniformly distributed across 10 object classes.

We apply standard data augmentation techniques to the training set, including random horizontal flipping followed by normalization with mean $[0.4914, 0.4822, 0.4465]$ and standard deviation $[0.2470, 0.2435, 0.2616]$. The test set is only normalized. The same augmentation strategy is applied to CIFAR-100, using mean $[0.5071, 0.4867, 0.4408]$ and standard deviation $[0.2673, 0.2564, 0.2762]$.

CNNs are trained on the original $32 \times 32$ resolution, while ViT models receive images resized to $224 \times 224$ through the data pipeline.

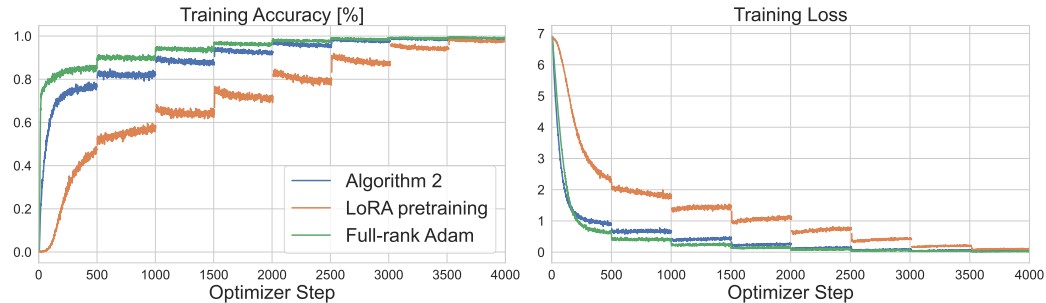

Figure 2: ViT-L.32 on ImageNet1k, pretrained from scratch in low-rank and full-rank baseline format for 4000 iterations. Training loss and accuracy of Algorithm 2 is close to the full-rank baseline, whereas LoRA pretraining struggles to converge within the training time budget.

### 10.1.4 ImageNet-1k Data

The ImageNet dataset consists of 1000 classes and over 1.2 million RGB training images, with a standard resolution of $224 \times 224$ pixels. We follow the standard data augmentation pipeline for ImageNet, which includes a random resized crop to $224 \times 224$, and normalization using mean $[0.5, 0.5, 0.5]$ and standard deviation $[0.5, 0.5, 0.5]$. The test set is only resized and center-cropped to $224 \times 224$, followed by normalization.

### 10.1.5 Additional Results - Transfer Learning

**ViT-L.32 on ImageNet1k**   We repeat the experimental setup on ViT-L.32 on the ImageNet-1k dataset, where ViT-L.32 is initialized with a Huggingface Imagenet21K checkpoint. We compare the baseline model, LoRA-based simultaneous descent pretraining, and Algorithm 2 in Table 7. We observe that Algorithm 2 is able to recover the baseline accuracy up to a small margin whereas LoRA-based training exhibits decreased Top-1 and Top-5 accuracy. Finally, we remark that the slightly lower compression rate is expected since the hidden dimension of ViT-L.32 (1024) is close to the number of ImageNet classes (1000), thus there is less redundancy in the model compared to other reported benchmarks.

Table 7: Results on ImageNet-1k with ViT-L.32 (304M parameters). Compression rate (c.r.) is reported in percent.

| Method | c.r. [%] | Top-1 Acc. [%] | Top-5 Acc. [%] |
|---|---|---|---|
| Baseline | 0 | **74.37** | 92.20 |
| Algorithm 2 | 61.45 | 72.27 | **90.19** |
| LoRA Pretrain | 60.00 | 63.20 | 84.81 |

### 10.1.6 Additional Results - Transfer Learning: Low-Rank Heavyball Method

**VGG16 on Cifar10**   We consider VGG16 on Cifar10 with Heavyball SGD using the same hyperparameters as described in Section 10.1.1. We choose $\gamma$=0.9 and train a (full-rank) baseline, LoRA pretrain and Algorithm 1. The compression rate of LoRA pretrain is fixed to match the final compression rate of Algorithm 1. In Table 8, we observe similar performance of Algorithm 1 to the Baseline as we saw for Algorithm 2 to the Adam Baseline, whereas LoRA pretrain exhibits a slight drop in accuracy.

Table 8: Results on Cifar10 with VGG16 using low-rank Heavyball SGD. Compression rate (c.r.) is reported in percent.

| Method | c.r. [%] | Acc. [%] |
|---|---|---|
| Baseline | 0 | 78.98 |
| Algorithm 1 | 94.35 | **79.01** |
| LoRA transfer learning | 93.72 | 75.12 |

### 10.1.7 Additional Results - Low-Rank Pretraining

**ViT-small on Cifar10**   We consider a compact Vision Transformer architecture for the CIFAR-10 dataset, see Section 10.1 for details on training and architecture. We compare baseline full-rank training with LoRA pretraining, Algorithm 2, and the naive implementation of Adam with DLRT [28]. Instead of low-rank finetuning, we pretrain the network from scratch and initialize the weights with a normal distribution. The low-rank methods factorize the fully-connected layers, while keeping the attention layers, which are typically high-rank, in baseline format. The LoRA rank is chosen to match the final compression rate of the rank adaptive

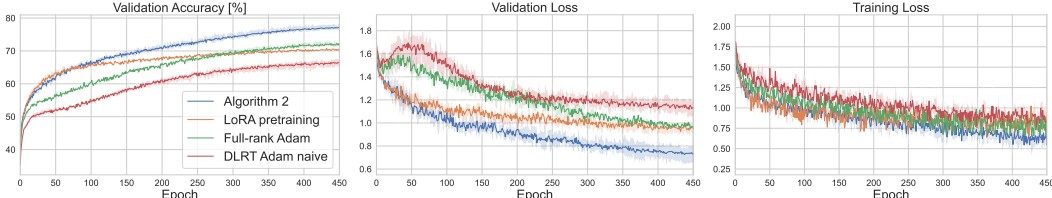

Figure 3: ViT-small on Cifar10, pretrained from scratch in low-rank and full-rank baseline format for 450 epochs. Median trajectory over 5 runs. Algorithm 2 and LoRA pretraining initially converge faster than the full-rank baseline. After the initial warm-up phase, Algorithm 2 exhibits a steeper convergence slope than LoRA. Moreover, Algorithm 2 achieves lower loss and higher validation accuracy than LoRA, even surpassing the baseline. A naive DLRT implementation with Adam leads to slower convergence and over 10% drop in validation accuracy.

naive DLRT and Algorithm 2 method, which achieves a compression rate of 67%. Remark that the compression rate is lower, since the attention matrices remain full-rank.

The goal of this test is to compare the long-term convergence behavior of all four methods, presented in Figure 3. We observe that Algorithm 2 and standard LoRA pretraining first converge faster than the baseline training. The non-orthogonal bases $A, B$ of LoRA and the corresponding non-orthogonal projection onto the low-rank manifold cause LoRA to plateau, whereas Algorithm 2 achieves lower loss values and higher validation accuracies, even surpassing the full-rank baseline in this test case. Naive implementation of DLRT with the Adam optimizer causes a more than 10% reduction in validation accuracy and slower convergence.

## 10.2 GLUE Benchmark

### 10.2.1 Dataset description

We present the benchmark overview in Table 9. We evaluate Algorithm 2 against several recent

Table 9: Summary of GLUE benchmark tasks

| Corpus | Task | #Train | #Dev | #Test | #Label | Metrics |
|--------|------|--------|------|-------|--------|---------|
| **Single-Sentence Classification (GLUE)** | | | | | | |
| CoLA | Acceptability | 8.5k | 1k | 1k | 2 | Matthews corr |
| SST2 | Sentiment | 67k | 872 | 1.8k | 2 | Accuracy |
| **Pairwise Text Classification (GLUE)** | | | | | | |
| MNLI | NLI | 393k | 20k | 20k | 3 | Accuracy |
| RTE | NLI | 2.5k | 276 | 3k | 2 | Accuracy |
| QQP | Paraphrase | 364k | 40k | 391k | 2 | F1 |
| MRPC | Paraphrase | 3.7k | 408 | 1.7k | 2 | Accuracy |
| QNLI | QA/NLI | 108k | 5.7k | 5.7k | 2 | Accuracy |
| **Text Similarity (GLUE)** | | | | | | |
| STS-B | Similarity | 7k | 1.5k | 1.4k | 1 | Pearson/Spearman cor |

finetuning methods on the General Language Understanding Evaluation (GLUE) benchmark [35]. GLUE is a standard benchmark comprising a diverse set of natural language understanding tasks that assess a model's ability to comprehend and process human language. It provides a broad evaluation by including tasks covering various linguistic aspects such as entailment, sentiment, and semantic similarity. The benchmark comprises the following nine tasks:

- **CoLA (Corpus of Linguistic Acceptability)**: Determines if a sentence is grammatically acceptable.
- **SST-2 (Stanford Sentiment Treebank)**: A binary sentiment classification task distinguishing between positive and negative sentiment.
- **MRPC (Microsoft Research Paraphrase Corpus)**: Identifies whether two given sentences are paraphrases.

- **STS-B (Semantic Textual Similarity Benchmark)**: Measures the semantic similarity of two sentences on a continuous scale from 1 to 5.
- **QQP (Quora Question Pairs)**: Assesses whether two questions are semantically equivalent.
- **QNLI (Question Natural Language Inference)**: Determines if a context sentence correctly answers a question.
- **RTE (Recognizing Textual Entailment)**: A binary entailment classification task.
- **Specific Focus:** MRPC (Microsoft Research Paraphrase Corpus)

The F1 score, used for evaluation, is computed from the precision $P$ and recall $R$ as follows. The precision $P$ is defined as

$$P := \frac{P_T}{P_T + P_F}, \tag{16}$$

where $P_T$ denotes the number of true positives and $P_F$ the number of false positives. The recall $R$ is given by

$$R := \frac{P_T}{P_T + N_F}, \tag{17}$$

where $N_F$ represents the number of false negatives. The F1 score is then the harmonic mean of $P$ and $R$:

$$F1 := \frac{2PR}{P + R}. \tag{18}$$

### 10.2.2 Reference implementations

**Full Finetuning (FT)**: The standard approach in transfer learning, where the model is initialized with pre-trained weights and all parameters are updated via gradient descent.

**Bitfit [39]**: Finetuning where only the bias terms are updated while all other parameters remain fixed.

**Adapter Tuning [14, 23]**: Involves inserting two-layer adapter modules within transformer blocks. In [14], adapters are placed between the self-attention and feed-forward modules with a residual connection (denoted HAdapter). In [23], adapters are inserted after the feed-forward and layer normalization modules (denoted PAdapter), following the notation of [42].

**LoRA [15]**: Applies low-rank additive updates to selected weight matrices, modeled as

$$\mathbf{z} = \sigma \left( W_{\mathrm{pt}} \mathbf{x} + \frac{\alpha}{r} AB^\top \mathbf{x} \right), \tag{19}$$

where $A, B \in \mathbb{R}^{n \times r}$. We apply LoRA to the attention matrices $W_q, W_k, W_v$, and the feed-forward matrices $W_{f_1}$ and $W_{f_2}$. Learning rates and optimizers follow the setup in [42], Appendix D–F.

Results for FT, Bitfit, Adapter tuning, and LoRA in Table 2 are reproduced from [42]. The performance of DoRA, LoRA, LoRA+, and AdaLoRA is computed using the HuggingFace implementations of these adapters.

**DoRA [21]**: A low-rank adapter similar in structure to LoRA, but with normalized $AB$ matrices and an additional magnitude parameter. Unlike LoRA, DoRA initializes the adapter with the pre-trained weights $W_0$, rather than zero.

**LoRA+ [10]**: Differs from LoRA in the assignment of learning rates: separate learning rates are used for $A$ and $B$, with a fixed ratio $\lambda_B / \lambda_A = 1.1$.

**AdaLoRA [42]**: Introduces adaptive low-rank updates to selected weight matrices:

$$\mathbf{z} = \sigma \left( W_{\mathrm{pt}} \mathbf{x} + \frac{\alpha}{r} USV^\top \mathbf{x} \right), \tag{20}$$

with frozen base weights $W_{\mathrm{pt}} \in \mathbb{R}^{n \times n}$, rank-$r$ adapters $U, V \in \mathbb{R}^{n \times r}$, and scaling matrix $S \in \mathbb{R}^{r \times r}$. The rank is determined using either SVD-based truncation or sensitivity analysis of the singular vectors. AdaLoRA is applied to $W_q, W_k, W_v, W_{f_1}$, and $W_{f_2}$ with an orthogonality regularization coefficient $\gamma = 0.1$.

When comparing to AdaLoRA, we align the total parameter budget with LoRA by setting the final budget $b^{(T)}$ to 576, and initialize with $b^{(0)} = 1.5 \times b^{(T)}$.

We also compare AdaLoRA using budget schedules obtained via Algorithm 2, ensuring that $b^{(T)}$ approximately matches the parameter count of the final models trained using Algorithm 2.

**GeoLoRA [27]**: GeoLoRA integrates the projected gradient flow Equation (9) in a parallelizable single-step scheme, including a rank adapative augmentation-truncation scheme as the proposed method. However, the method is only applicable for stochastic gradient descent, and not yet extended to momentum-based approaches. We use the hyperparameter choices reported in [27].

We use the implementation of [42, Appendix C] to compute the results for the presented reference methods. We set the exponential moving average parameters $\beta_1$ and $\beta_2$ of AdamW as their pytorch default value. We select the learning rates as denoted in Table 10, selected by an initial hyperparameter sweep.

We implement Algorithm 2 s similar as possible to the reference models to achieve a fair comparison. That is, we add an adapter of the form $\mathbf{z} = \sigma(W_{\text{pt}}\mathbf{x} + USV^\top\mathbf{x})$ to the key $W_k$, query $W_q$ and value $W_v$ matrices of all attention blocks, and to both feed-forward layers $W_{f_1}$ and $W_{f_2}$. For each adapter, we employ Algorithm 2 to update the layer weights and ranks.

Table 10: Hyper-parameter setup for the GLUE benchmark, determined by an initial hyperparameter sweep.

| Dataset | Learning Rate | Batch Size | # Epochs | $\tau$ | init. rank | Adapter dropout | weight decay |
|---------|---------------|------------|----------|--------|------------|-----------------|--------------|
| RTE | $1.2 \times 10^{-3}$ | 32 | 20 | 0.075 | 10 | 0.01 | 0.01 |
| QNLI | $5 \times 10^{-4}$ | 64 | 5 | 0.05 | 10 | 0.2 | 0.01 |
| MRPC | $1 \times 10^{-4}$ | 64 | 5 | 0.05 | 10 | 0.15 | 0.05 |
| QQP | $1 \times 10^{-4}$ | 64 | 5 | 0.05 | 10 | 0.15 | 0.05 |
| SST-2 | $1 \times 10^{-4}$ | 64 | 10 | 0.05 | 10 | 0.05 | 0.01 |
| CoLA | $5 \times 10^{-4}$ | 32 | 25 | 0.05 | 10 | 0.1 | 0.01 |
| STS-B | $1 \times 10^{-3}$ | 128 | 30 | 0.05 | 10 | 0.05 | 0.1 |

## 10.3 Llama2 7b-chat-hf on BoolQ and PIQA

**BoolQ** is a reading comprehension dataset consisting of naturally occurring yes/no questions paired with passages from Wikipedia. Questions are drawn from real Google search queries, and each is annotated with an answer by human raters, making it a benchmark for natural, open-domain question answering.

**PIQA** (Paragraph-level In-context QA) is a dataset designed for evaluating in-context learning in long-form reading comprehension. It provides paragraph-length passages with associated questions and answers, emphasizing models' ability to extract relevant information from extended contexts rather than isolated sentences.

Table 11: Hyper-parameter setup for Algorithm 2 for the reasoning benchmark Table 3, determined by an initial hyperparameter sweep.

| Dataset | Learning Rate | Batch Size | # Epochs | $\tau$ | init. rank | Adapter dropout | weight decay |
|---------|---------------|------------|----------|--------|------------|-----------------|--------------|
| BoolQ | $1.76 \times e^{-4}$ | 12 | 3 | 0.0696 | 6 | 0 | 0.1 |
| PIQA | $1.36 \times e^{-4}$ | 12 | 3 | 0.0838 | 6 | 0 | 0.1 |

Table 12: Hyper-parameter setup for LoRA for the reasoning benchmark Table 3, determined by an initial hyperparameter sweep.

| Dataset | Learning Rate | Batch Size | # Epochs | $\tau$ | init. rank | Adapter dropout | weight decay |
|---------|---------------|------------|----------|--------|------------|-----------------|--------------|
| BoolQ | $4.47 \times e^{-4}$ / $1.76 \times e^{-4}$ | 12 | 3 | None | 6/10 | 0 | 0.1 |
| PIQA | $2.04 \times e^{-4}$ / $1.36 \times e^{-4}$ | 12 | 3 | None | 6/10 | 0 | 0.1 |

## 10.4 GPT2 on OpenWebText

**OpenWebText** is an open-source dataset constructed as a replication of OpenAI's WebText. It was created by scraping URLs shared on Reddit. The dataset contains web pages spanning diverse topics,

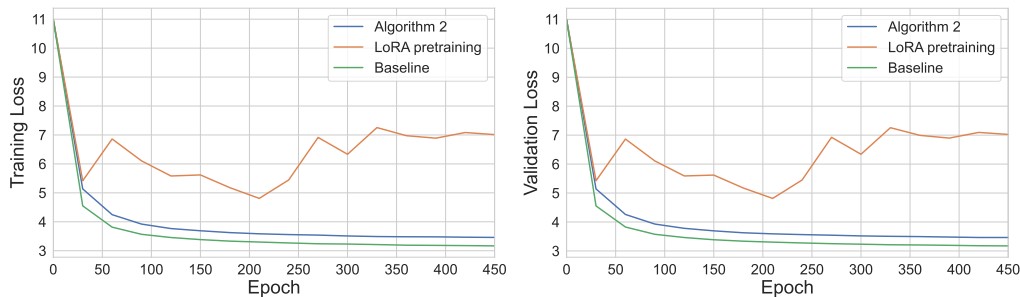

Figure 4: GPT2 reproduction on OpenWebText, pretrained from scratch in low-rank, full-rank baseline and Algorithm 2 for 15000 iterations. Algorithm 2 method significantly outperforms LoRA pretraining (best validation loss $3.4642$ vs. $4.8141$), while incurring only a moderate increase relative to the full-rank baseline ($3.4642$ vs. $3.2313$).

filtered to remove duplicates and non-English text, and is commonly used as a large-scale corpus for training and evaluating language models.

Table 13: Hyperparameter configuration for pretraining GPT-2 (124M) on OpenWebText (see Table 4).

| Dataset | Learning Rate | Batch Size | # iteration | $\tau$ | init. rank | Adapter dropout | weight decay |
|---|---|---|---|---|---|---|---|
| OpenWebText | $[6e^{-4}, 6e^{-5}]$ | 64 | 15 000 | 0.05 | 135 | 0 | 0.1 |

## 10.5 Computational hardware

All experiments in this paper are computed using workstation GPUs. Each training run used a single GPU, except for GPT-2 pretraining, which was performed on two NVIDIA H100 GPUs. Specifically, we have used 5 NVIDIA RTX A6000, 3 NVIDIA RTX 4090, and 2 NVIDIA H100.

