# OpenReview forum: "A geometric framework for momentum-based optimizers for low-rank training"
_NeurIPS.cc/2025/Conference — NeurIPS 2025 poster_

### Official Review · Reviewer_9kiS · 2025-06-30

**Clarity:** 3
**Significance:** 3
**Originality:** 3
**Rating:** 5
**Confidence:** 3

**Summary:**

This paper introduces a new momentum based optimization framework for low rank pre-training and fine-tuning. The proposed method combines dynamic low-rank approximation and momentum methods. Compared to existing layer factorization method DLRT, the proposed method extend the applicable optimizers from SGD to Heave-ball momentum gradient descent and Adam.

The proposed framework is supported with theoretical convergence and error bound guarantees, the authors also provide extensive experimental results on pre-training, fine-tuning and transfer learning to demonstrate the performance of proposed method generally matches the full rank baseline.

**Questions:**

1. This paper mainly provides experimental results for Algorithm 2, how does the performance of Algorithm 2 compare to that of Algorithm 1?
2. In the experiment "Pretraining from scratch of ViT-small on Cifar10", algorithm 2 exhibits faster convergence and smaller validation loss compared to the full rank baseline. Could the authors provide some insights or explanations for this observed behavior?

**Ethical Concerns:**

["NO or VERY MINOR ethics concerns only"]

**Final Justification:**

The authors have addressed my questions, my rating remains

**Limitations:**

Yes

**Paper Formatting Concerns:**

No formatting concerns

**Quality:**

4

**Strengths And Weaknesses:**

Strengths
1. This paper extends DLRT to momentum optimizers, including both Heavy-ball momentum gradient descent and Adam, bridging an existing gap in the literature.
2. The authors provide a comprehensive treatment of the proposed framework, including detailed derivations, clear computational complexity and memory analysis, as well as theoretical convergence and error bound guarantees.
3.  This paper is well structured and easy to follow

Weaknesses
1. To enhance the readability of the paper for general readers who may not be familiar with LoRA or DLRT, it would be beneficial to include more explanations of the benchmark methods used in the experiment results, adding algorithmic descriptions for DLRT w/o proj and LoRA pretrain

---

> ### Author Rebuttal · Authors · 2025-07-30
>
> We thank the reviewer for their constructive comments and appreciate that they found the paper comprehensive, well structured and that it bridges a gap in the literature.
>
>
> 1. (W1) We fully agree and have added the Algorithmic descriptions for both methods in the Appendix in the revised manuscript and will upload the version as soon as the review process allows it.
>
>     For completeness we point out the differences of the methods below.
>
>      - DLRT w/o projection does not perform the projection update of $S_{\mathcal{V}}$ in Line 7 and 12 of Algorithm 3 in the supplementary material. Thus it creates an inconsistency of the coefficient representation $S$ and the corresponding optimizer states, since they are now spanned by different bases.
>      - LoRA pretrain essentially applies Algorithm 4 to the LoRA factors $A,B\in\mathbb{R}^{n\times r}$, where $r$ is fixed, as stated in Line 244.
>
> 2. (Q1) The Dynamical Low-Rank Adam method is the main result of the paper, as the (full rank) Adam optimizer is currently the most widely used training method for neural networks - to the best of our knowledge. Thus we focused our evaluation on this setting.
>
>     For completeness we added VGG16 on Cifar10 with SGD using the same hyperparameters as in the experiments in table 1. We choose momentum=0.9 and train a (full-rank) baseline, LoRA pretrain and Algorithm 1. The compression rate of LoRA pretrain is fixed to match the final compression rate of Algorithm 1. We observe similar  performance of Algorithm 1 to the Baseline as we saw for Algorithm 2 to the Adam Baseline, whereas LoRA pretrain exhibits a slight drop in accuracy.
>
> | **Method**         | **c.r. [%]** | **Accuracy [%]** |
> |----------------|------------------------|--------------|
> | Baseline       | 0.00                   | 78.98        |
> | Algorithm 1        | 94.35                  | 79.01        |
> | LoRA Pretrain  | 93.72                  | 75.12        |
>
> 3. (Q2) In our experience, and as investigated in recent literature [a], Vision Transformers train relatively slow on small datasets, e.g. Cifar10, and exhibit worse performance than ViT models pretrained on large datasets (ImageNet size). We think that encouraging low-rank solutions by training with Algorithm 2 helps eleviate this issue to some extend - explained by an Occam's Razor argument: A low-rank solution expresses a certain simplicity bias imposed on the very general transformer architecture.
>     One can see that the low-rank initialization leads to a fast loss decay in Figure 2b, whereas the validation loss of the full vision transformer stalls first. Interestingly the LoRA loss, also with a low-rank initialization decays fast at first, but then stalls. This might be attributed to the missing convergence guarantees to a low-rank optimum (see Section 2).
>
>     A similar effect is observed in the GLUE benchmark portfolio, where low-rank finetuning methods (LoRA-based and DLRT-based) outperform full-fine tuning of certain sub-benchmarks.
>
>
>     To the best of our knowledge, this often seen observation is currently not rigorously understood and future research may shed more light onto this phenomenon.
>
>
>     [a] H. Zhu, B. Chen, and C. Yang. Understanding why vit trains badly on small datasets: An intuitive perspective, 2023.

---

> ### Comment · Reviewer_9kiS · 2025-08-02
>
> I thank the authors for addressing my questions and detailed explanation, my rating remains

---

### Official Review · Reviewer_yZDw · 2025-06-30

**Clarity:** 2
**Significance:** 3
**Originality:** 4
**Rating:** 4
**Confidence:** 4

**Summary:**

This paper studies the momentum-based algorithm when applying to low-rank weight optimization in neural network training. The paper identifies a condition where the heaby-ball method converges to the low-rank optimum solution, and shows that directly applying momentum to the factorization form of the weight does not satisfy this condition. Based on this condition, the paper derives the heavy-ball and Adam method for low-rank problems. The paper presents experimental results in the main text, and a convergence guarantee in the appendix to corroborate the effectiveness of the proposed algorithm.

**Questions:**

Does the proposed algorithm also work for tasks like matrix completion and matrix sensing?

**Ethical Concerns:**

["NO or VERY MINOR ethics concerns only"]

**Final Justification:**

The rebuttal resolves most of my concerns. The only thing that is kind of weak from my perspective is the theoretical convergence guarantee of the algorithm. Despite the empirical choice of $nu$ that could lead to a meaningful convergence result, the paper did not provide a strong enough convergence guarantee.

**Limitations:**

Yes.

**Paper Formatting Concerns:**

None.

**Quality:**

3

**Strengths And Weaknesses:**

**Strength**

1. The algorithm design comes from a solid theoretical principle and looks quite convincing.

2. Per experimental result, the performance is better than the chosen baseline for a significant amount.

**Weaknesses**

1. The motivation of using the proposed algorithm is a bit weak. The paper points out the issue with directly applying momentum to DLRT by showing that the direct application does not satisfy (4). However, in the appendix it is only shown that (4) is a sufficient condition, but it is not clear whether (4) is also a necessary condition. Therefore, I believe that the explanation of why direct application of the momentum to DLRT is inconclusive.

2. I am not sure whether the experimental result in Table 1,2 are measured in epochs or in total training time. If the performance is based on a fixed number of epochs, then the comparison can be unfair as the proposed algorithm imposes a constant times of additional per-iteration cost.

3. Convergence guarantee in Theorem 3 is not strong enough. In particular, Theorem 3 bounds the difference between the algorithm trajectory and an ideal trajectory that converges to the low-rank optimum. However, the upper bound depends on $\hat{\nu}$, which is the gap between the projection of the momentum onto the current basis and the augmented basis. Intuitively, the augmented basis should explore more possibility, so this difference can be quite large.

---

> ### Author Rebuttal · Authors · 2025-07-30
>
> We wish to thank the reviewer for their thoughtful and constructive feedback. We are encouraged that the reviewer found our algorithm design to
> be based on a solid theoretical principle and that our experimental results
> show significant performance improvements.
> The reviewer raised three main points regarding the motivation, experimental setup, and the interpretation of our convergence guarantee. We
> address each weakness (W) and question (Q) below.
>
> ---
>
> 1. **W1 (On the motivation for the proposed gradient flow):**
>
>     The reviewer correctly points out that we showed that satisfying equation (4) is a *sufficient* condition for convergence to a low-rank optimum, but we did not explicitly demonstrate that naive momentum methods can fail to converge to a point fulfilling $P(W)\nabla_W \mathcal{L}(W)=0$. We thank the reviewer for highlighting this, as a more concrete discussion improves the paper's motivation.
>
>     The potential lack of convergence for a naive application of momentum methods can be proven with a simple but illustrative counter-example. Consider a state where the momentum term is zero, i.e., $\mathcal{V}=0$, by choosing the momentum factors as $U_v = -U$, $S_v = 0$, and $V_v = V$. Now, consider any weight matrix $W$ that is *not* a low-rank optimum, meaning $P(W)\nabla_W \mathcal{L}(W) \neq 0$, but for which the naive update term happens to be zero: $\hat{P}(W,\mathcal{V})\nabla_W \mathcal{L}(W) = 0$.
>
>     A large class of matrices fulfills this. For instance, any $W=USV^\top$ where the gradient is non-zero in the range of V but zero in the range of U (i.e., $U^\top\nabla_W \mathcal{L}(W) = 0$ but $\nabla_W \mathcal{L}(W)V \neq 0$) would satisfy this. This can be easily verified: Since $\mathcal{V}=0$, we have
>
> $$
> \hat{P}(W,\mathcal{V})\nabla_W \mathcal{L}(W) = UU^\top\nabla_W \mathcal{L}(W)VV^\top = 0,
> $$
>
> but
>
> $$
> P(W)\nabla_W \mathcal{L}(W) = \nabla_W \mathcal{L}(W)VV^{\top}\neq 0.
> $$
>
> In this scenario, the naive gradient flow equations would identify $(W, \mathcal{V})$ as a stationary point, since $\dot{W}=0$ and $\dot{\mathcal{V}}=0$. However, this point is not a valid low-rank optimum because the true projected gradient $P(W)\nabla_W \mathcal{L}(W)$ is non-zero. In contrast, $(W, \mathcal{V})$ is not a stationary point of (4) and the gradient flow (4) will provably drive the system to a state where $P(W)\nabla_W \mathcal{L}(W)=0$.
>
> This demonstrates a fundamental geometric inconsistency in the naive approach: it can prematurely halt at suboptimal points that do not correspond to true low-rank critical points. **We added this counter-example and discussion to the appendix of the revised manuscript to make our motivation clearer and more compelling.**
>
>
>
> 2. **W2 (On the experimental setting and runtime comparison):**
>
>     The details for the numerical experiments are provided in the Appendix in Table 3 and 5, respectively. We set an epoch limit for all experiments in the paper, where we have chosen the epoch number such that all methods have sufficient iterations to converge. This is a default test case setup.
>
>
>     To underline that Algorithm 2 does not require a significantly increased training time, we provide a wall clock comparison of finetuning  Llama2 7b-chat-hf on BoolQ and PICA benchmarks
>     with standard LoRA [Hu et al., 2021] and Algorithm 2 in the table below. The wall-time is reported for three epochs with batch size 12 and maximal sequence length of 640 tokens on a single NVIDIA H100. We observe singificant outperformance of Algorithm 2 at a negligible wall-time overhead for all test cases.
>
> | **Method**              | **BoolQ Acc [%]** | **Wall-Time** | **Params** | **PIQA Acc [%]** | **Wall-Time** | **Params** |
> |----------------------------|-------------------|---------------|------------|------------------|---------------|------------|
> | Algorithm 2                | 84.09%            | 186min        | 0.270%     | 76.77%           | 228min        | 0.247%     |
> | LoRA (r=6)                 | 62.17%            | 173min        | 0.179%     | 52.18%           | 225min        | 0.179%     |
> | LoRA (r=10)                | 62.17%            | 184min        | 0.299%     | 50.43%           | 225min        | 0.299%     |
>
>
>
> Hyperparameters are determined by initial hyperparameter sweeps (which were performed for LoRA and Algorithm 2 individually). Hyperparameters for Algorithm 2 are reported below:
>
> | **Dataset** | **Learning Rate** | **Batch Size** | **# Epochs** | **τ**    | **Init. Rank** | **Adapter Dropout** | **Weight Decay** |
> |---------|----------------|------------|-----------|--------|-------------|------------------|---------------|
> | BoolQ   | 1.76e-4         | 12         | 3         | 0.0696 | 6           | 0                | 0.1           |
> | PIQA    | 1.36e-4         | 12         | 3         | 0.0838 | 6           | 0                | 0.1           |
> ---
>
> Hyperparameters for LoRA:
>
> | **Dataset** | **Learning Rate** | **Batch Size** | **# Epochs** | **τ**    | **Init. Rank** | **Adapter Dropout** | **Weight Decay** |
> |---------|---------------|------------|----------|------|-------------|------------------|---------------|
> | BoolQ   | 4.47e-4 / 1.76e-4      | 12         | 3        | None | 6 / 10      | 0                | 0.1           |
> | PIQA    | 2.04e-4 / 1.36e-4      | 12         | 3        | None | 6 / 10      | 0                | 0.1           |
>
> 2. **W3 (On the derived error bound):**
>
>     We appreciate the reviewer's detailed question regarding Theorem 3. We wish to note that our algorithm can be straightforwardly modified to provably control the term $\widehat \vartheta$ (as underlined in the comment following Theorem 3 in our manuscript in Line 431). In this case, $\widehat{\vartheta}$ becomes a user-determined parameter. We chose to present the main algorithm without this explicit modification because our analysis and empirical results suggest it is unnecessary and adds computational overhead for no tangible benefit. To name an example, we obtain 84.15% (instead of 84.09%) validation accuracy when finetuning Llama2 7b-chat-hf on the BoolQ benchmark when controlling the projection term by choosing $\widehat{\vartheta}=0.1$, which is fulfilled automatically for almost all iterations.
>
>     Intuitively, the influence of this term is naturally limited, as it is expected to diminish during training. This expectation is based on the dynamics of the optimization process: as the optimizer converges to a local minimum, the momentum term $\mathcal{V}$ itself must converge toward zero.
>     This is a direct consequence of Theorem 1 in the Appendix.
>     Consequently, the difference in its projection onto two closely related subspaces will also naturally tend to zero. This intuition is strongly backed by the consistent high performance of our method across all experiments, where this term was not explicitly controlled.
>
>     To perhaps clarify the context of our response, we feel it's helpful to underline the role of Theorem 3 within our work. We included this analysis in the appendix to provide readers with theoretical intuition for our algorithmic design—specifically, to show how our integrator navigates the numerical instability associated with inverting the $S_v$ matrix. It serves as a motivational stepping stone for our final algorithm, which is the paper's primary contribution. Our goal is a novel, efficient method derived from a geometric framework, and the ultimate validation for such a method lies in its empirical performance. We believe the current algorithm, which, as the reviewer noted, shows significant performance improvements, is the strongest version to present, as it achieves these results at minimal cost without unneeded operations.
>
>     We will revise the discussion around Theorem 3 in the appendix to better articulate this interplay between theoretical motivation and practical algorithm design.
>
>
> 3. **Q1 (Applicability to matrix completion/sensing):**
>
>     Absolutely. While our paper focuses on neural network training, the proposed geometric framework is general. The core of our method is a momentum-based optimizer for problems with low-rank constraints. Applying it to classic low-rank optimization problems like matrix completion and matrix sensing is a promising and natural extension. We hypothesize that our approach could offer similar benefits in convergence speed and stability for these tasks, particularly in challenging, ill-conditioned scenarios.
>
>     We remark that we do not aim for a state-of-the-art matrix completion algorithm as there exists a rich corpus of specialized methods that are tailored to the specific structure of a matrix completion problem and can therefore outperform more general methods in this specific task.
>
>
>
> We hope these clarifications, along with the planned additions to the appendix, fully address the reviewer's concerns. We are confident that these improvements will strengthen the paper and hope the reviewer will reconsider their rating.

---

> > ### Comment · Reviewer_yZDw · 2025-08-03
> >
> > Thank you for your detailed response and additional experimental results. I believe that my concerns has been resolved, and I will raise the score accordingly.
> >
> > One thing that I would respectfully disagree with the author is that I believe that it is important to have a convergence guarantee for the proposed algorithm, despite a strong motivation for where the algorithm is derived from. However, I also recognized that the empirical choice of $\nu = 0.1$ indeed would not cause too much trouble for the convergence guarantee.

---

### Official Review · Reviewer_Q6vZ · 2025-07-01

**Clarity:** 3
**Significance:** 3
**Originality:** 3
**Rating:** 5
**Confidence:** 3

**Summary:**

Momentum-based updates for low-rank training needs to be geometry-aware for correctness. The paper shows how naive implementation of LoRA with heavy ball method (independently updating each of the factors in Equation 3) does not ensure that update dynamics stay on the low-rank manifold. The paper gives a framework for the evolution of the factorized weights $W$ and factorized momentum terms $\mathcal{V}$. Using this framework, the paper provides update steps and algorithms for two methods: low-rank heavy ball method and low-rank Adam method. The paper also addresses two numerical challenges in their methods: evolving $W$ in factorized form to maintain low-cost and avoiding stiffness in the dynamical system through the avoidance of inverting $S_v$ and $S_v^\intercal$. Experimental results show that the proposed Algorithm 2 (low-rank Adam) gives both higher accuracy and higher compression rate than existing methods and naive implementation of low-rank momentum.

**Questions:**

- Do the convergence results given in Theorems 1 and 2 for continuous-time dynamics carry over to discrete algorithms (such as Algorithms 1 and 2)?
- Is there an extra e in Lady Winderemere's fan on line 430?

**Ethical Concerns:**

["NO or VERY MINOR ethics concerns only"]

**Final Justification:**

Because the additional experimental results are preliminary and because the theoretical results in the paper does not necessarily characterize the proposed algorithms, I will maintain my rating at 5 (Accept).

**Limitations:**

Yes

**Quality:**

3

**Strengths And Weaknesses:**

Strengths:
 - The paper addresses a short-coming of other LoRA-style methods: the updates may leave the low-rank manifold. It does so by developing a rigorous framework for momentum-based optimization methods on the manifold of low-rank matrices.
 - Proposes low-rank versions of two popular methods: Adam and Heavy Ball. The algorithms keeps inner product calculations only on low-ranked factorized matrices and thus is efficient.
 - Empirical results look good: the proposed method achieves high accuracy with high compression rates.

Weaknesses
 - The experiments do not show any wall-clock time measurements. Although theoretically, all operations on done on much smaller factorized matrices, but even small operations (e.g. the truncation step in Algorithm 2) could have significant cost in the aggregate.

---

> ### Author Rebuttal · Authors · 2025-07-30
>
> We thank the reviewer for their insightful comments.
>
> 1. **Wall-clock Timing (W1):**
>
>     We fully agree that it was unclear how additional operations increase the overall runtime compared to standard LoRA training.  Therefore, we decided to provide a wall clock comparison of a bigger benchmark to demonstrate that aditional operations do not significantly increase the runtime. We have added wall clock comparisons for finetuning Llama2 7b-chat-hf on BoolQ and PICA benchmarks with standard LoRA [Hu et al., 2021] and Algorithm 2 in the table below. The wall-time is reported for three epochs with batch size 12 and maximal sequence length of 640 tokens on a single NVIDIA H100. We observe singificant outperformance of Algorithm 2 at a negligible wall-time overhead for all test cases.
>
> | **Method**              | **BoolQ Acc [%]** | **Wall-Time** | **Params** | **PIQA Acc [%]** | **Wall-Time** | **Params** |
> |----------------------------|-------------------|---------------|------------|------------------|---------------|------------|
> | Algorithm 2                | 84.09%            | 186min        | 0.270%     | 76.77%           | 228min        | 0.247%     |
> | LoRA (r=6)                 | 62.17%            | 173min        | 0.179%     | 52.18%           | 225min        | 0.179%     |
> | LoRA (r=10)                | 62.17%            | 184min        | 0.299%     | 50.43%           | 225min        | 0.299%     |
>
> Hyperparameters are determined by initial hyperparameter sweeps (which were performed for LoRA and Algorithm 2 individually). Hyperparameters for Algorithm 2 are reported below:
>
> | **Dataset** | **Learning Rate** | **Batch Size** | **# Epochs** | **τ**      | **Init. Rank** | **Adapter Dropout** | **Weight Decay** |
> |---------|----------------|------------|-----------|--------|-------------|------------------|---------------|
> | BoolQ   | 1.76e-4         | 12         | 3         | 0.0696 | 6           | 0                | 0.1           |
> | PIQA    | 1.36e-4         | 12         | 3         | 0.0838 | 6           | 0                | 0.1           |
>
> Hyperparameters for LoRA:
>
> | **Dataset** | **Learning Rate** | **Batch Size** | **# Epochs** | **τ**      | **Init. Rank** | **Adapter Dropout** | **Weight Decay** |
> |---------|---------------|------------|----------|------|-------------|------------------|---------------|
> | BoolQ   | 4.47e-4 / 1.76e-4      | 12         | 3        | None | 6 / 10      | 0                | 0.1           |
> | PIQA    | 2.04e-4 / 1.36e-4      | 12         | 3        | None | 6 / 10      | 0                | 0.1           |
>
> 2. **Convergence Results Transfer (Q1):**
>    The convergence result for Theorem 2 directly carries over to the discrete setting of Algorithm 1. The reason for this is that a first order time discretization will not add error terms that are asymptotically bigger than the current error rates.
> The situation of Algorithm 2 become much more challenging, since there does not exist a direct interpretation of ADAM as a time dis-
> cretization of a gradient flow. We point this out in Section 4, Line 185. Theorem 1 does not directly translate into a discrete setting and a discrete version would, most probably, require additional conditions on the learning rate.
>
> 3. **Typo Fix (Q2):**
>    You're right — it should be *Windermere*. We've corrected it in the revised manuscript.

---

> > ### Author Response · Authors · 2025-08-05
> >
> > Dear reviewer,
> >
> > as encouraged by the email by the program chairs, we wanted to use this opportunity to reach out.
> >
> > We hope the additional wall-clock benchmarks and clarifications regarding the convergence results have fully addressed your questions. Thank you again for your thoughtful review and for helping us improve the paper.

---

> > ### Comment · Reviewer_Q6vZ · 2025-08-07
> >
> > I thank the authors for their response, especially the additional experimental results with wall-clock times reported and the clarification on whether the convergence results of the theorems carry over to their respective algorithms used in practice. Because the additional experimental results are preliminary and because the theoretical results in the paper does not necessarily characterize the proposed algorithms, I will maintain my rating at 5 (Accept).

---

### Official Review · Reviewer_sz9N · 2025-07-01

**Clarity:** 2
**Significance:** 4
**Originality:** 3
**Rating:** 4
**Confidence:** 3

**Summary:**

This paper theoretically demonstrates that conventional momentum-based optimizers like Heavy Ball and Adam fail to converge to local optima in low-rank neural network training due to ignoring the geometric constraints of the low-rank manifold's tangent space. To address this, it extends the Dynamic Low-Rank Training (DLRT) theory and propose a geometry-aware optimization framework that explicitly enforces momentum updates to remain within the tangent space. It achieves 95% parameter compression in both image classification (pretraining) and natural language processing benchmarks (finetuning) with comparable performance.

**Questions:**

Mentioned in Strengths And Weaknesses.

**Ethical Concerns:**

["NO or VERY MINOR ethics concerns only"]

**Final Justification:**

After rebuttal and discussion, the authors clarified the review’s concerns, addressed typographical issues, and committed to improving equation referencing, which resolved several presentation concerns. The paper provides valuable and novel insight into the limitations of momentum-based optimizers and demonstrates impressive compression performance with competitive accuracy, highlighting its potential impact in both theory and practice. While the empirical validation is currently limited to small datasets without large-scale experiments, and real-world wall-clock training times, direct convergence rate comparisons, and more accessible theorem explanations would further strengthen the work, these limitations appear addressable in future revisions. Overall, the contributions are significant and the results compelling, making this a promising paper that is well worth borderline acceptance.

**Limitations:**

Mentioned in Strengths And Weaknesses.

**Quality:**

3

**Strengths And Weaknesses:**

Strengths
- Theoretical Analysis of Limitations: The paper theoretically demonstrates that conventional momentum-based optimizers fail to converge to low-rank optima due to geometric constraints.
- Memory Efficiency with Performance: Achieves comparable accuracy to full-rank baselines at huge compression ratio across multiple benchmarks (image domain and NLP).

Weakness
- Type (Line 67): `optimizer` should be corrected to `optimizer`.
- Clarity on gamma in equations: I think that gamma represents the damping coefficient but there is no explicit definition.
- Ambiguous Reference to equations: It’s better to use expressions such as Eq. (5) or Equation (5) rather than (5). It makes use be confused
- Theorem Placement: Theorems are briefly mentioned in the main text without explanation; integrating key theorems into the main body could improve readability.
- Missing Wall Time Comparison: The paper analyzes the computational efficiency, but real-world training time (wall clock time) is absent.
- Theoretical Convergence Rate: A direct convergence rate comparison with existing methods helps the claim of faster convergence mentioned in the abstract.
- ImageNet Pretraining: Experiments are conducted on smaller datasets (CIFAR, UCM), but validation on large-scale datasets like ImageNet is missing. Pretraining, in this context, refers to training on large datasets to enhance generalization performance. Training on large-scale datasets like ImageNet could strengthen the persuasiveness of the paper.
- The overall notation explanation needs to be improved for better clarity and readability.

---

> ### Author Rebuttal · Authors · 2025-07-30
>
> Thank you very much for the encouraging feedback and for your useful
> comments and constructive suggestions. We believe the reviewer’s comments and suggestions have helped greatly in improving our manuscript.
> We will upload the revised version as soon as the review process allows.
>
> 1. We have corrected that typo.
>
> 2. $\gamma$ denotes the momentum decay parameter. A clarifying remark has been added after Eq. (3) in the revised manuscript, and Section 7 in the appendix now includes a table with all notation definitions.
>
> 3. All occurrences of *(num)* have been replaced by *Eq. (num)* in the revised manuscript.
>
> 4. We chose to included theorems in the appendix to provide interested
> readers with theoretical intuition for our algorithmic design while not
> hindering readability of our work. However, our main contribution
> is the statement of Algorithm 2 and its empirically demonstrated
> performance. If you believe it increases readability to directly state
> theorems in the main paper, we are happy to include these.
>
> 5. We fully agree that it was unclear how additional operations increase the overall runtime compared to standard LoRA training. Therefore, we decide to provide a wall clock comparison of a bigger benchmark to demonstrate that aditional operations do not significantly increase the runtime. We provide a wall clock comparison of finetuning  Llama2 7b-chat-hf on BoolQ and PICA benchmarks
> with standard LoRA [Hu et al., 2021] and Algorithm 2 in the table below. The wall-time is reported for three epochs with batch size 12 and maximal sequence length of 640 tokens on a single NVIDIA H100. We observe singificant outperformance of Algorithm 2 at a negligible wall-time overhead for all test cases.
>
> | **Method**              | **BoolQ Acc [%]** | **Wall-Time** | **Params** | **PIQA Acc [%]** | **Wall-Time** | **Params** |
> |----------------------------|-------------------|---------------|------------|------------------|---------------|------------|
> | Algorithm 2                | 84.09%            | 186min        | 0.270%     | 76.77%           | 228min        | 0.247%     |
> | LoRA (r=6)                 | 62.17%            | 173min        | 0.179%     | 52.18%           | 225min        | 0.179%     |
> | LoRA (r=10)                | 62.17%            | 184min        | 0.299%     | 50.43%           | 225min        | 0.299%     |
>
> Hyperparameters are determined by initial hyperparameter sweeps (which were performed for LoRA and Algorithm 2 individually). Hyperparameters for Algorithm 2 are reported below:
>
> | **Dataset** | **Learning Rate** | **Batch Size** | **# Epochs** | **τ**      | **Init. Rank** | **Adapter Dropout** | **Weight Decay** |
> |---------|----------------|------------|-----------|--------|-------------|------------------|---------------|
> | BoolQ   | 1.76e-4         | 12         | 3         | 0.0696 | 6           | 0                | 0.1           |
> | PIQA    | 1.36e-4         | 12         | 3         | 0.0838 | 6           | 0                | 0.1           |
>
> Hyperparameters for LoRA:
>
> | **Dataset** | **Learning Rate** | **Batch Size** | **# Epochs** | **τ**      | **Init. Rank** | **Adapter Dropout** | **Weight Decay** |
> |---------|---------------|------------|----------|------|-------------|------------------|---------------|
> | BoolQ   | 4.47e-4 / 1.76e-4      | 12         | 3        | None | 6 / 10      | 0                | 0.1           |
> | PIQA    | 2.04e-4 / 1.36e-4      | 12         | 3        | None | 6 / 10      | 0                | 0.1           |
>
>
> 6. First, we remark that we only make claims about the empirical convergence rate.
> To that end, we have provided a convergence rate comparison
> in Figure 2, where we train a ViT model on Cifar-10 from scratch with
>
>     (a) full-rank (baseline) Adam (Algorithm 4),
>
>     (b)  the LoRA-based Adam, i.e. Algorithm 4 applied to all factors of the low-rank factorization individually,
>
>     (c) the proposed Adam-DLRT (Algorithm 2), and
>
>     (d) a naive implementation of Adam for DLRT, that does not change the mo-
> mentum states.
>
> As observed in Figure 2, the proposed Algorithm 2
> has a fast initial convergence, and a higher final accuracy than the
> comparison methods. For empirical soundness, we have added the
> standard deviation to the image in the revised manuscript.
>
> 7. We provide numerical results for the ImageNet-1k (1.2 Million im-
> ages), where we compress the ViT-L32 vision transformer (304 Million
> parameters). We first conduct a hyperparameter sweep to determine
> standard hyperparameters as learning rate, weight-decay and batch
> size: ViT-L32 is trained on ImageNet with a batch size of 256, learning rate 0.001, for 20 epochs using AdamW optimizer and L2 regularization of 0.0001; DLRT uses a relative truncation tolerance of 0.013 and initial rank 200.
>
>     We compare then the baseline model, LoRA-based simulateneous descent pretrianing, and Algorithm 2  in the table below.
> We observe that Algorithm 2 is able to recover the baseline accuracy up to a small margin whereas LoRA pretraining exhibits decreased top1 and top5 accuracy.
>
>
>    | **Method**        | **c.r. [%]** | **Top-1 Acc. [%]** | **Top-5 Acc. [%]** |
>    |---------------|----------|----------------|----------------|
>    | Baseline      | 0        | 74.37          | 92.20          |
>    | Algorithm 2   | 61.45    | 72.27          | 90.19          |
>    | LoRA Pretrain | 60.00    | 63.20          | 84.81          |
>
>    Finally we wish to remark that the slightly lower compression rate is expected since the hidden dimensions of ViT-L32 of 1024 is close to the number of ImageNet classes (1000), thus there is less redundancy in the model compared to other reported benchmarks.
>
>
> 8. We unified notation across momentum and Adam equations, using consistent symbols for the first moment terms in the revised manuscript. A Notation table is now included in the appendix. We believe that this makes it much easier to follow our manuscript and we thank the reviewer for pointing this out.

---

> > ### Comment · Reviewer_sz9N · 2025-08-06
> >
> > Thanks for your responses!
> > While many concerns have been partially addressed, making claims about the empirical convergence rate would still seem to require a broader range of experiments. Alternatively, a theoretical analysis of the convergence rate could be a valuable addition, though it may require some time. Additionally, while the compression limitations of ViT-L32 is discussed for ImageNet, it would be helpful to explore whether similar compression limitations exist in other widely-used Transformer architectures and datasets. Do these models (along with datasets) also exhibit such compression constraints, potentially limiting the general applicability and effectiveness of the proposed method?

---

> > > ### Author Response · Authors · 2025-08-07
> > >
> > > We thank the reviewer for their reply and their clarifying questions.
> > >
> > > 1. (in response to "making claims about the empirical convergence rate would still seem to require a broader range of experiments") As stated in our abstract, "We validate our methods through numerical experiments, demonstrating faster convergence, and stronger validation metrics at given parameter budgets." The part "faster convergence [...] at given parameter budgets" is intended to highlight our empirical observation that our method converged faster than other optimizers in our experiments relative to their parameter count. We hope that this does not give the impression that we claim a universally applicable convergence rate accross all benchmarks.
> > >
> > > You've raised a fair point that our initial discussion was primarily focused on Figure 2. To provide broader and more detailed support, we will update the manuscript to:
> > >
> > > a) Add a direct comparison of LoRA and Algorithm 2 in the discussion of Figure 2, highlighting that our method exhibits a steeper slope after the initial warmup phase.
> > >
> > > b) Add additional convergence plots for our other key experiments to the appendix. As an example, the table below shows the training accuracy history for ViT-L32 on ImageNet, where Algorithm 2 again converges faster than LoRA-Pretrain.
> > >
> > > | iter | Baseline | Algorithm 2 | LoRA-Pretrain
> > > |---- | ---- | ---- | ----|
> > > | 0   |0.001953125|0.0015625| 0.00078125|
> > > 500   |0.8578125|0.862890625|0.473046875 |
> > > 1000  |0.89140625 |0.88671875|0.573046875 |
> > > 1500  |0.9390625|0.90560803|0.6234375|
> > > 2000  |0.98515625|0.9703125|0.719921875|
> > > 2500  |0.989453125|0.984765625|0.806640625|
> > > 3000  |0.991015625|0.991796875|0.869140625|
> > > 3500  |0.994921875|0.990234375|0.94140625
> > >
> > > See also the convergence history in our response regarding point 2 (below), where we show faster convergence of Algorithm 2 compared to LoRA-Pretrain.
> > >
> > > Moreover, we fully agree with you that deriving theoretical convergence rates is infeasible (because this cannot be done in the limited time remaining in this review phase, and also given that such proofs are still lacking for the original (full-rank) Adam optimizer on non-convex objectives).

---

> > > > ### Author Response · Authors · 2025-08-07
> > > >
> > > > 2. (in response to "it would be helpful to explore whether similar compression limitations exist in other widely-used Transformer architectures and datasets. Do these models (along with datasets) also exhibit such compression constraints, ...")
> > > > Yes, in general, the achievable compression rate of any low-rank training method is closely tied to the inherent complexity of the task and dataset and compression becomes a trade-off between memory footprint and performance metrics. As a result, while the network loses accuracy to some extend, the resulting memory footprint is reduced and finding a good trade off is highly user and problem dependent. To further explore such scenarios, we pretrained GPT2 (124 Million parameters, reproducing Karpathy's GPT-2 implementation) from scratch on the OpenWebText (9 Billion training tokens and 4.4 Million validation tokens) benchmark, i.e. next-word prediction. In this situation, we are able to significantly outperform LoRA pretraining (best validation loss of 3.4642 vs. 4.8141), while exhibiting a moderate increase in validation loss compared to the full-rank baseline (3.4642 vs. 3.2313). We achieve a compression rate of 39.39% for Algorithm 2 and 39.21% for LoRA-Pretrain. Thus, our method is capable to allow for compressing GPT-2 (leading to the potential to substantially reduced inference time) while LoRA leads to a significant increase in validation loss.
> > > >
> > > > | iter | Baseline | Algorithm 2 | LoRA-Pretrain
> > > > |---- | ----      | ---- | ----|
> > > > | 0   |10.9906 | 10.9906 | 10.9903
> > > > 1 000  |4.6510 | 5.1414 | 5.4255
> > > > 2 000  |3.9026 | 4.2600 | 6.8642
> > > > 3 000  |3.6478 | 3.9279 | 6.1070
> > > > 4 000  |3.5291 | 3.7806 | 5.5900
> > > > 5 000  |3.4572 | 3.6942 | 5.6200
> > > > 6 000  |3.4022 | 3.6274 | 5.1751
> > > > 7 000  |3.3594 | 3.5902 | 4.8141
> > > > 8 000  |3.3342 | 3.5646 | 5.4503
> > > > 9 000  |3.3150 | 3.5437 | 6.9251
> > > > 10 000 |3.2950 | 3.5189 | 6.3392
> > > > 11 000 |3.2772 | 3.5048 | 7.2568
> > > > 12 000 |3.2617 | 3.4956 | 6.9926
> > > > 13 000 |3.2515 | 3.4797 | 6.8977
> > > > 14 000 |3.2402 | 3.4648 | 7.0940
> > > > 15 000 |3.2313 | 3.4642 | 7.0242
> > > >
> > > > Since we did not have time to sweep hyperparameters, we simply choose the default parameters that were reported for the Baseline in the Karpathy repository, for Algorithm 2 and LoRA-Pretrain. We remark that this scenario gives an advantage to the Baseline.
> > > >
> > > > To put the results in context, we deliberately chose a low-rank pretraining benchmark for which we expected moderate compression rates and would like to remind the reviewer that in LoRA fine-tuning and LoRA-based transfer learning, compression rates exceeding 90% are routinely achieved (especially for widely-used Transformer architectures and datasets).
> > > >
> > > > 3. (in response to "... potentially limiting the general applicability and effectiveness of the proposed method")
> > > > Low-rank training methods have a wide applicability and effectiveness accross a large number of application fields, see e.g., the remarkable success of LoRA (currently at > 17,000 citations since its introduction in 2022). There certainly exist applications in which compression becomes a trade-off between memory footprint and accuracy. The situation described is best viewed as an accuracy-vs-compression trade-off, which is an inherent characteristic of compression in certain challenging scenarios.
> > > > In our work, we showcase a wide range of numerical experiments from diverse application fields that include large datasets and network architectures. These experiments show significantly improved network performance compared to conventional low-rank methods such as LoRA, including e.g., DeBERTaV3-base on GLUE, LLama2 on BoolQ and PIQA, ViTL-32 on ImageNet, and more. These results are not intended to advertise low-rank methods as a general replacement of full-rank training, but highlight that our method is highly effective compared to other successful low-rank training methods.
> > > > To make the nature of the trade-off clearer to the reader, we are happy to add the following sentence to our manuscript:
> > > > "It is important to note that the accuracy-vs-compression trade-off varies by application. While low-rank methods excel in fine-tuning and transfer learning tasks (sometimes even improving upon the baseline), pre-training a network from scratch on a complex dataset often involves balancing memory savings against a potential drop in accuracy."

---

> ### Author Response · Authors · 2025-08-05
>
> Dear reviewer,
>
> as encouraged by the email by the program chairs, we wanted to use this opportunity to initiate a discussion.
> We believe we have addressed your concerns in our answer above.
> Specifically, we have provided extensive additional numerical experiments that showcase the **viability and scalability of our method** on
>
>    a) compression of models on large datasets (ViT-L32 vision on ImageNet) and
>
>    b) large model finetuning (Llama2 7b-chat-hf on BoolQ and PICA)
>
> We hope this fully addresses your concerns. Should any questions remain, we would of course be happy to clarify further.
>
> Thank you again for your thoughtful review and for helping us improve the paper.

---

### Note · Authors · 2025-08-12

We thank the reviewers for their insightful feedback, which has helped us strengthen our paper. The reviewers highlighted numerous strengths, including the paper's **solid theoretical foundation** (sz9N,Q6vZ,9kiS,yZDw) and the method's **strong empirical performance on diverse benchmarks** (sz9N,Q6vZ,9kiS,yZDw). Our paper **bridges an existing gap** in the literature (9kiS) and is **well structured and easy to follow** (9kiS).

We'd like to use the final remarks to summarize how we addressed weaknesses and discussion points:

- **Wall-Time Comparisons** (sz9N,Q6vZ,yZDw): To address concerns about overhead, we've added wall-time results (Llama2-7B), demonstrating that our method's additional operations do not lead to significantly longer training times. In this new experiment, additional operations in our method lead to **only 1% wall time overhead over LoRA, at 24% accuracy gain**.

- **Empirical Convergence** (sz9N): To provide a broader discussion of the empirically observed faster convergence, we have added new convergence data (ViT-L32 on ImageNet, GPT2 on OpenWebText) and included the convergence plots for our other key experiments in the appendix. In the new results, our method's convergence rates are **close to the full-rank baseline**, and **outperform classical LoRA training by a large margin**.

- **Compression Rates** (sz9N): We clarified that the accuracy/compression trade-off is **inherent to all low-rank training methods**, especially for low-rank pretraining from scratch. Crucially, we demonstrated that **our method significantly mitigates this trade-off**. We substantiated this with major new pretraining experiments (ViT-L32 on ImageNet-1k, GPT-2 on OpenWebText).

- **Additional analysis** (yZDw): 1) We added a proof that conventional **low-rank gradient flows can fail to converge to an optimal point**. 2) We added a **justification backed by numerical experiments** for the assumption in Theorem 3 and additionally underlined that it can be **explicitly enforced through a minor modification** as already mentioned in the original manuscript.

- **Improved Clarity** (9kiS,sz9N): We added a table summarizing our notation and added pseudo-codes for DLRT without projection and LoRA pretrain.

All points of the reviewers have been addressed through additional large-scale experiments (Llama2-7b, GPT2, ViT-L32), deeper discussions, and further analysis. Finally, we sincerely thank all reviewers and the AC for their time and constructive engagement.

---

### Decision · Program_Chairs · 2025-09-17

**Decision:**

Accept (poster)

**Comment:**

This paper proposes a method for low rank training of neural networks, an important problem in the fine-tuning of foundation models. For the most part the reviewers praised both the theoretical and the empirical results, with the main concerns being about clarity and wallclock improvements. Given that these have been addressed with proposed modifications and new results, I recommend acceptance.